# Question-based computational language approach outperforms rating scales in quantifying emotional states
Sverker Sikström ⊠, Ieva Valavičiūtė, Inari Kuusela & Nicole Evors

Psychological constructs are commonly quantified with closed-ended rating scales. However, recent advancements in natural language processing (NLP) enable the quantification of open-ended language responses. Here we demonstrate that descriptive word responses analyzed using NLP show higher accuracy in categorizing emotional states compared to traditional rating scales. One group of participants ($N = 297$) generated narratives related to depression, anxiety, satisfaction, or harmony, summarized them with five descriptive words, and rated them using rating scales. Another group ($N = 434$) evaluated these narratives (with descriptive words and rating scales) from the author's perspective. The descriptive words were quantified using NLP, and machine learning was used to categorize the responses into the corresponding emotional states. The results showed a significantly higher number of accurate categorizations of the narratives based on descriptive words (64%) than on rating scales (44%), questioning the notion that rating scales are more precise in measuring emotional states than language-based measures.

While the use of artificial intelligence (AI) in mental health is a promising area of research, it is essential to ensure that the algorithms being developed are reliable, accurate, and transparent. The accuracy of AI-based mental-health-related technology can be estimated by comparing the AI-based methods for analyzing language with ground truth, where common methods are standardized rating scales. In this work, we take on another approach by introducing a paradigm where participants in Phase 1 are instructed to write an autobiographical narrative of an emotional state that is read by participants in Phase 2. This approach is interesting as the validation is directly generated by the participants' self-experienced emotions. We probe participants with a single open-ended language question related to narrative emotions where they respond with five descriptive words that are analyzed by AI methods and compare these with standardized rating scales.

Language is a natural way for people to communicate their mental states. Nonetheless, standardized numeric rating scales are the dominating way that behavioral scientists measure mental states, as they are believed to have higher validity. For example, in a typical research article featured in the Journal of Personality and Social Psychology, 87% of the data used to derive conclusions are based on closed-ended rating scales[1]. However, these scales have limitations as they are one-dimensional, typically ranging from "strongly disagree" to "strongly agree", they tend to produce an error of central tendency, have a halo effect, or are limited by the capacity for self-observation, to name a few. Scholars are becoming increasingly skeptical

regarding the validity of rating scales because they may risk oversimplifying the human experience[2,3].

Alternatively, open-ended language responses may convey a more person-centered and holistic representation of one's mental state. For instance, there may be multiple ways in which indicators of a specific disorder could be expressed, addressing symptoms other than those prescribed by the DSM-V[4]. In instances of major depressive disorder, additional signs such as somatic symptoms (e.g., headaches or digestive issues) may be present, which are not commonly linked with depression. DSM-V is commonly critiqued for its undue emphasis on categorizing symptoms based on specific criteria, potentially overlooking the diverse presentations and variations within mental disorders[5]. It also receives criticism for being too rigid, potentially excluding individuals with clinically significant symptoms that do not meet the specified criteria[6]. This suggests that adopting an approach with an open-ended response format may allow for greater flexibility in the diagnostic process, which can accommodate individual variations and contextual factors. An open-ended response could offer a more comprehensive assessment of mental health conditions. This approach may complement the diagnostic manual, enhancing the accuracy and personalized understanding of mental health conditions. By considering a wider range of indicators beyond those prescribed by the DSM-V, researchers and clinicians can gain a more nuanced understanding of mental health conditions that

Department of Psychology, Lund University, Lund SE-221 00, Sweden. ⊠e-mail: sverker.sikstrom@psy.lu.se

may better capture the complexity of each individual and allow for a more personalized treatment plan.

Manual analysis of open-ended responses has drawbacks: it is a time-consuming and effort-intensive process susceptible to personal biases[7]. Consequently, prior to the boom of computer technology, the widespread adoption of rating scales emerged as a favored alternative, alleviating these challenges. Recent findings indicate that the traditional ways in which mental illnesses are captured (e.g., the PHQ-9 for depression or the GAD-7 for anxiety) might be overlooking other, equally important symptoms associated with a specific mental illness[8]. Fortunately, recent advancements in natural language processing (NLP) offer a potential solution for efficiently interpreting and quantifying language responses while maintaining test-retest reliability.

Technological progress, especially in AI and machine learning (ML), has profoundly improved the accuracy and efficiency of predicting outcomes by analyzing large datasets and identifying patterns that may not be readily apparent through traditional methods (e.g., basic statistical modeling, or simple regression analyses). Such processes resulted in the automation of decision-making across diverse domains, as the system autonomously learns tasks without external intervention or supervision. Recent developments in AI have simplified the work of marketing professionals, clinicians, statisticians, and various analysts, and promising results have been observed in fields like web searches[9,10], targeted marketing[11], and finance[12]. These ML applications have resulted in the development of sentiment analyzers, text classifiers, chatbots, and virtual assistants that may be capable of transforming the field of mental health diagnostics as well[8].

ML models have shown great promise in assessing and predicting mental disorders based on widely different datasets, for genetics, magnetic resonance imaging, electroencephalography, and clinical data (e.g., type 2 diabetes)[13]. In clinical settings, a popular application of ML has been NLP, which has been used especially in electronic health records[14,15], medical diagnosis[16] and social media text data mining[17]. Nevertheless, using such input data necessitates having access to, and being able to provide, a substantial amount of up-to-date medical records. To overcome these limitations, we focus on data that can readily be assessed by prompting participants to answer a single open-ended question related to mental health. Such responses can be answered in seconds by a few descriptive words and compared with commonly used rating scales, each consisting of a number of items and several response alternatives.

The field of mental health has witnessed an accelerated application of AI, contributing to the improvement of diagnostic and treatment methodologies. Private corporations such as Amazon have already made considerable progress in this field. For instance, Amazon has developed a patent that allows its Alexa device to identify depression and suicidal tendencies[18] and plans to combine this technology with its healthcare and pharma businesses, creating new opportunities for profit. However, the development of algorithms for mental health by private companies may not always be transparent. There are concerns about the potential biases and ethical implications of these algorithms, particularly when they are developed without appropriate oversight and regulation[19]. Here, research is essential to ensure that the algorithms are reliable and valid. For instance, there have been advancements in NLP algorithms that can analyze patient conversations and identify patterns that suggest potential clinical issues[20]. A lot of work within the field has been focused on developing computational models that can predict mental health outcomes from social media data. For example, it has been shown that it is possible to identify individuals who are at risk for depression or anxiety by analyzing their social media posts[21–23]. Although social media studies have shown great promise, assessment of the present emotions requires current and relevant social media data, which is not always accessible.

In recent years, significant advancements have been observed in NLP models. The emergence of the transformer, a powerful ML technique, has been associated with remarkable performance improvements. Among them, BERT (Bidirectional Encoder Representations from Transformers) stands out as the most frequently cited transformer-based language model[24].

Transformers are flexible and sizable statistical models renowned for their ability to capture word meanings in context. Assessments consistently demonstrate that BERT achieves substantial error reductions compared to earlier models[24]. With their large size and flexibility, these models excel at representing diverse word meanings in different contexts, enhancing researchers' capability to grasp the nuanced intent of speakers and writers. While it is true that BERT comes with its limitations, including subjectivity and systemic biases that may reflect diagnostic and social stereotypes[25], it has undeniably made a significant breakthrough in the field of NLP. BERT's pioneering contribution lies in its introduction of unsupervised pre-trained models, allowing them to draw insights from extensive sets of unlabeled text data[26].

Studies have shown that text-based answers analyzed by computational methods can indeed predict corresponding close-ended rating scales such as PANAS, Ryff's Scales of Psychological Well-being, the Satisfaction with Life Scale (SWLS), Depression Anxiety and Stress Scales, and others. The concept of word tagging has been recently introduced for retrieving emotional states through an Emotional Recall Task, where participants listed 10 emotional states experienced in the last month[27]. The study demonstrated a significant correlation between the Emotional Recall Task and PANAS, shedding light on diverse mental search strategies, especially in individuals with varying positive or negative affect levels[27]. Building on this, researchers developed DASentimental, a semi-supervised ML model grounded in the Emotional Recall Task[28]. This innovative system learned to map psychometric measurements of anxiety, stress, and depression to user-generated word sequences, employing search strategies rooted in cognitive science and associative knowledge modeling. Question-based Computational Language Assessment (QCLA) involves generating text by asking participants to answer open-ended questions that can be transformed into a quantifiable vector using NLP[29]. This method captures a severity measure of mental health and a detailed description of mental states. A related study also points out the validity of QCLAs where human cooperation is distinguished more precisely using computational language assessment, analyzed using NLP, than by using traditional rating scales[30]. Collectively, these studies emphasize the importance of cognitive information that can be derived from word-based responses using AI. This is particularly relevant for clinical researchers using, for example, unstructured clinical notes that are otherwise undocumented and carry no added benefits.

While NLP approaches offer advantages, past research on language-based methods has not achieved the same level of accuracy as rating scales. The reason for this is that the previous validation of language-based approaches has primarily used rating scales as the outcome measure, where validity is measured by a correlation to the rating scales and therefore does not allow testing of whether rating scales or language measures have the highest validity[29,31]. To address this issue, the present study compares descriptive word-based responses and rating scales using an outcome criterion that is independent of rating scales. In particular, we focused on an outcome criterion where participants were instructed to generate self-experienced narratives with specific emotional content related to key psychological constructs either depression, anxiety, satisfaction, or harmony (also referred to as emotional states). The narratives produced by the participants were then evaluated by others using both rating scales and descriptive words. We investigated how well these evaluations can categorize narratives into the emotions used to generate the narratives.

The primary and original contribution of this study lies in its examination of the accuracy of assessing emotional states using person-generated text responses compared to rating scales. While computational methods for analyzing language data have been previously introduced in other works[29], this study focuses specifically on the comparative evaluation of these two assessment approaches. The present work aims to show that language-based measures on emotional states, analyzed by NLP, have higher accuracy in categorizing emotional states compared to commonly used rating scales dedicated to measuring these states. In particular, we prompted participants with a single open-ended question to which they responded with five descriptive words (i.e., "Write five keywords that best capture your/ the

**Fig. 1 | Paradigm design.** The figure shows the paradigm used in the study. In phase 1, on the left-hand side, participants are instructed to write an autobiographical narrative of an episode where they experienced either depression, anxiety, satisfaction, or harmony. Following this, they summarize the emotions in the narrative with five descriptive words and fills in rating scales corresponding to the four emotions (i.e., PHQ-9, GAD-7, SWILS, and HILS respectively). The five words are quantified by a large language model (BERT) into a vector. A machine learning algorithm (multiple logistic regression) is used to categorize either the vector or the total score of the rating scales. This categorization is compared with the emotion that participants were instructed to use while writing the narrative. Phase 2 is identical to Phase 1, with the exception that the participants read the stories generated in Phase 1 (rather than writing them).

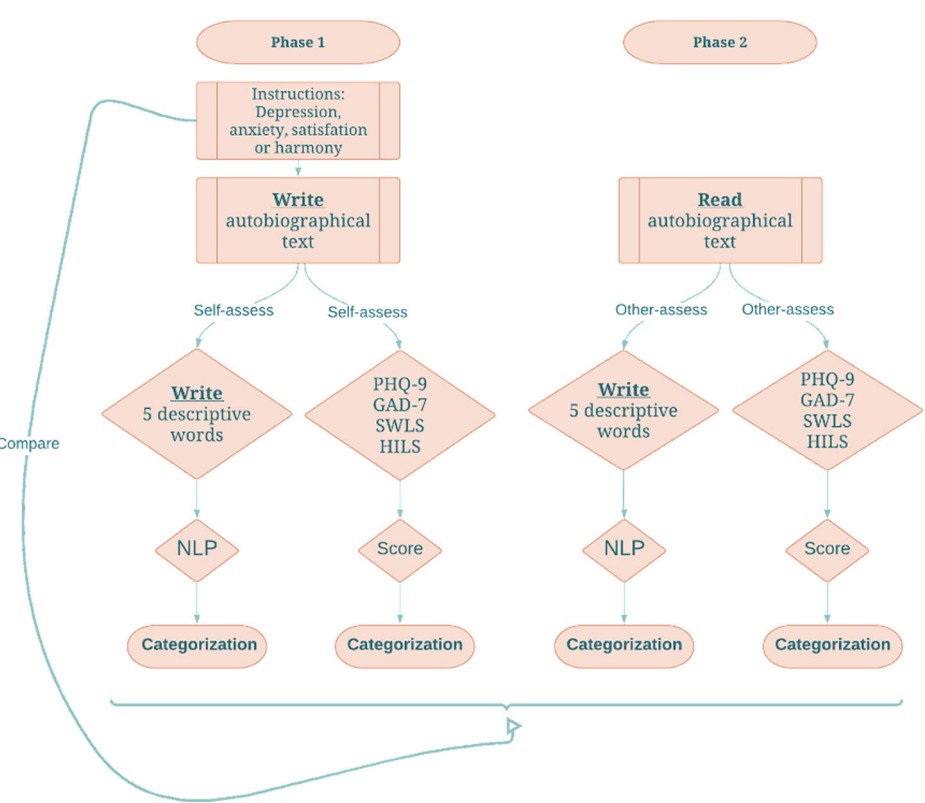

author's emotional state"). We chose five descriptive words as the response format as previous work shows that this data type shows more accurate predictions compared to free text data and adding more words does not significantly increase the predictive accuracy[29]. Writing a few descriptive words can be done in a reasonably short time (e.g., seconds) while writing a free text narrative typically takes longer (e.g., minutes). We also selected some of the most common psychological concepts relating to emotional states, namely, depression, anxiety, satisfaction, and harmony in life, because they represent a blend of positive psychology and clinical psychology elements, each characterized by a specific emotional valence that can be quantified through the use of standardized rating scales and are typically highly correlated[30,32]. The selection of these four psychological constructs also aligns with previous works from our research group, ensuring consistency and facilitating the comparability of findings across studies[30,33,34].

Here we introduced a new paradigm consisting of two phases, where the participants generate emotional narratives in the first phase that are assessed by other participants in the second phase (see Fig. 1). In Phase 1 of the study, participants were asked to write an autobiographical text about a specific self-experienced event of either depression, anxiety, satisfaction, or harmony. In Phase 2, a different set of participants (composed of controls and healthcare professionals) were asked to read a text written by someone in Phase 1. In both phases, participants described the emotional content of the narratives in five descriptive words, and completed rating scales corresponding to the four emotions (i.e., the Patient Health Questionnaire (PHQ-9), the Generalized Anxiety Disorder Scale (GAD-7), the SWLS, and the Harmony in Life Scales (HILS)).

We employed NLP techniques to convert the five-word responses into an embedding, specifically a semantic vector. This involved utilizing computational methods based on our earlier research work[29,30]. The classification of embeddings and rating scales was based on multinomial regression assigning each to one of the four emotional states. As an NLP model, we opted for a state-of-the-art transformer-based language model (i.e., BERT). The evaluation of the classification accuracy was based on a ten-fold cross-validation procedure. The pre-registered hypotheses

anticipated that in the evaluation of predetermined emotional states, descriptive word responses would demonstrate superior predictive capabilities compared to corresponding rating scales when subjected to NLP analysis (H1). Furthermore, it was hypothesized that the joint consideration of descriptive word responses and rating scale assessments for diverse emotional states would yield greater predictive accuracy than the assessment of each independently (H2). We also hypothesized that descriptive word responses matched on narratives show higher inter-rater reliability between Phase 1 and 2, compared to corresponding rating scales (H3). Finally, as the number of healthcare professionals was expected to be rather small, no specific hypotheses were made regarding this group. As such, we tested whether professionals' assessments of emotional states were more strongly associated with rating scales, indicating higher predictive capability, compared to controls (H4).

## Methods

The study received ethical approval from the Regional Ethics Board in Lund, Sweden, adhered to Swedish laws (Dnr 2021 – 04627), and the data collection was completed in May 2022. All methods were carried out in accordance with relevant guidelines and regulations, adhering to the Declaration of Helsinki. Written informed consent was obtained from all subjects prior to the start of data collection. The study's design, hypotheses, and analysis plan were preregistered on the Open Science Framework (OSF) (2022-03-15) before completion of the study, accessible at https://osf.io/6fx72. The precise wording of the third preregistered hypothesis on OSF was: "Semantic scales correlate to a higher extent than corresponding estimation scales when measured between the participants."

### Participants

The data used here were collected by convenience sampling. Participants were recruited through Prolific, an online recruitment platform for data collection in the behavioral sciences. The inclusion criteria were 18 years of age or older and being a native English language speaker. The study had two phases with different sets of participants but with the same inclusion criteria. The participants were compensated £2 in Phase 1 and £1.5 in Phase 2 for

**Table 1 | Participant demographic data**

| Phase | N Total[a] | N excluded | Age[b] | Gender | Nationality | Time[c] | Education[d] |
|---|---|---|---|---|---|---|---|
| 1 | 350 (297) | 53 | 19–76, 29.37 (13.62) | women-165<br>men-122<br>undisclosed-10 | US-115<br>UK-108<br>other-74 | 14.25 (17.51) | 107<br>62<br>46<br>6<br>76 |
| 2 | 465 (434) | 31 | 18–79, 34.57 (11.79) | women-263<br>men-159<br>undisclosed-12 | US-235<br>UK-110<br>other-89 | 9.84 (11.78) | 127<br>192<br>63<br>8<br>44 |
| 1 & 2 | 815 (731) | 84 | 18–79, 31.97 (12.71) | women-428<br>men-281<br>undisclosed-22 | US-350<br>UK-218<br>other-163 | 12.05 (14.65) | 234<br>254<br>109<br>14<br>120 |

[a] Total number of participants before after (parenthesized) the exclusion.
[b] Age is expressed by a minimum and a maximum age range, mean, and standard deviation in years;
[c] The amount of time spent completing the study is expressed by mean time (SD) in minutes and seconds;
[d] Level of education (from top to bottom): high school, undergraduate, postgraduate, Ph.D., other.

participating in the study. A total of 350 participants completed the study in Phase 1, and 53 participants were removed due to failure to correctly answer the control questions, leaving a final sample size of 297. A total of 465 participants completed the study in Phase 2, 34 of whom were healthcare professionals recruited through additional screening. The same platform was used to recruit healthcare professionals by screening participants based on their indication of having a professional occupation within the healthcare system as either; a doctor, emergency medical employee, nurse, paramedic, pharmacist, psychologist, or social worker. A total of 31 participants were removed as they did not respond correctly to the control questions, leaving a final sample size of 434. Thus, the final sample from both phases consisted of 731 participants (women = 428; men = 281; respondents with undisclosed gender = 22) with an age range of 18–79 years ($M = 31.97$, $SD = 12.71$) (see Table 1 for more details).

**Measures**

**Autobiographical narrative.** The study included an emotional auto-biographical text response, a descriptive-word response, four rating scales, and demographic questions. Participants in Phase 1 were asked to write about a self-experienced event when they felt either depression, anxiety, satisfaction, or harmony. The allocation of psychological constructs for participants to write about was done randomly, ensuring that each construct received an approximately equal number of responses. The instructions were formulated as follows: "Please write a text about a period in life (days to months) when you experienced [depression/anxiety/satisfaction/harmony]. Please answer the question by writing at least a paragraph (approximately five sentences). Write about those aspects that are the most important and meaningful to you. Note. Please do not use the word '[depression/anxiety/satisfaction/harmony]' in your text."

While these two pairwise psychological concepts (depression-anxiety and satisfaction-harmony) are theoretically distinct, they often exhibit high correlations with each other when assessed through rating scales. In light of this conceptual and criteria-based distinction, semantic measures in our study are suggested as a means to differentiate between them more distinctly than traditional rating scales.

**Descriptive words.** Participants were also asked to capture the emotional aspects of the narrative in five descriptive words (phrases containing more than one word were not allowed and participants were not permitted to proceed in the survey if more than one word was entered in the text field). The instruction in Phase 1 was the following: "Write five keywords that best capture the emotional state you wrote about earlier. Note. Please do not use the word [depression/anxiety/satisfaction/

harmony] in your text." Participants in Phase 2 were only asked to describe the mental state that they read about in the text without knowing which emotion, in particular, the text was referring to. Participants were presented with the texts in a random order, ensuring that each text was read at least once, but no more than twice. The phrasing for Phase 2 was the following: "Read this text about a period in someone's life and write five keywords that best capture the author's emotional state: [the text was inserted here]."

**Rating scales.** The following four standardized rating scales were used in this study. The PHQ-9 is a widely used tool to assess depression severity[35]. It consists of nine questions that correspond to the criteria for diagnosing major depressive disorder in the DSM-IV. The nine items on the PHQ-9 cover various symptoms of depression, such as mood, sleep, appetite, and energy levels. Each item is scored on a scale from 0 to 3. Participants' responses to the nine items are then added together to obtain a total score, ranging from 0 to 27. The total score indicates the severity of depression: minimal, mild, moderate, moderately severe, and severe depression. The Generalized Anxiety Disorder (GAD-7) scale is a commonly used mental health measure of the severity of generalized anxiety disorder[36]. The seven items on the GAD-7 are scored on a scale from 0 to 3. Participants' responses to the seven items are then added together to obtain a total score, ranging from 0 to 21. The total score indicates the severity of generalized anxiety disorder: minimal, mild, moderate, and severe anxiety. The SWILS is a widely used tool to measure an individual's overall life satisfaction[2]. The HILS measures a person's overall satisfaction and contentment with life[37]. Both HILS and SWILS scales consist of five statements that participants respond to on a scale from 1 to 7. Participants' responses to these items are then summed up to obtain a total score, which can range from 5 to 35. Higher scores indicate higher levels of life satisfaction/harmony in life, while lower scores suggest lower satisfaction/harmony in life.

Participants were asked to capture the emotional state of the narrative using the above-described scales. In other words, the instructions for the standardized rating scales were modified in Phase 1 so that they referred to emotions in the self-experienced event and not the present-day state, while in Phase 2 they were modified to correspond to the emotions in the read narrative (for details see the pre-registered report at OSF). In Phase 1, the rating scale instructions were as follows: "Over that period in life, how often have you been bothered by: ….". In Phase 2, the instructions were modified to be related to the read narrative, for example: "Consider the author's emotional state: Over that period of their life, indicate their agreement with each item by tapping the appropriate box.".

## Procedure

Participants were directed from Prolific, a platform for recruiting participants for research studies online, to a Qualtrics questionnaire, in which their responses were recorded. Upon the start of the survey on Qualtrics, a security technology reCAPTCHA was used to to distinguish between human users and automated bots. Having passed that stage, participants were informed about the purpose of the study, their right to withdraw at any time, and their responses being anonymous and voluntary. They were also told that no personal or identifiable information was being collected and that they could contact the researchers with any questions regarding the survey (for details see the pre-registered report at OSF).

In Phase 1 the participants were asked to write one autobiographical text about a period in their lives when they experienced one of the following emotional states: depression, anxiety, harmony, or satisfaction. Thus, each participant wrote one narrative about one emotion and was not informed about the other three emotions. The described emotional states were evenly distributed among the participants. Participants were then asked to write down five single words describing the emotional state of the narrative, complete the four standardized rating scales, and answer demographic questions that included age, gender, country of birth, and level of completed education.

The procedure for Phase 2 was identical to Phase 1, except that the participants were asked to read a text written by someone in Phase 1 rather than to write one themselves. Having read the text, they wrote five descriptive words about the narrative they read and completed the rating scales related to the emotions in the narratives, i.e., not their self-experienced emotions as in Phase 1 but those of the author. This was done by replacing "you" with "the author" and "they" in the wording of the semantic questions and rating scales. Each participant only read and responded to one narrative. However, because there were more participants in Phase 2 than in Phase 1, some of the narratives from Phase 1 were read and evaluated more than once in Phase 2 (but no more than twice).

In both phases, participants were given the freedom to write the narratives and complete the rest of the questionnaire for as long as they desired. The narratives underwent a manual review to guarantee the exclusion of any nonsensical data and no narratives had to be removed as a result. For the narratives, there was no minimum or maximum number of words. On average, participants wrote $N = 81.32$ words ($SD = 40.66$) in their narratives. Finally, to ensure that the participants were attentively following the instructions, in both phases four control questions were embedded, one among each of the rating scale questions. Participants were instructed to respond to a question with a pre-given alternative, for example: "Answer 'Agree' to this question". If any one of the four control questions was not answered correctly, that participant's data were removed from the final sample.

## Data analysis

The descriptive words generated by the participants were quantified using BERT. The predicted categorization of the four emotions was generated by multinomial logistic regression. The analyses were conducted in SemanticExcel.com[38], an online tool for statistical analysis of semantic data, where the main author's underlying code is written in MATLAB.

**Pre-processing of the semantic data.** Descriptive word and autobiographical narrative responses, collected as part of this study, were altered per the procedures provided by Kjell and colleagues[29]. The descriptive word responses generated by the participants were first cleaned using a manual procedure. Misspelled words were corrected by the spelling tool in Microsoft Word, and this was done only when the author's intended meaning was clear; otherwise, their original form was retained. Instances of incorrectly answered control questions, successively repeated words, or where participants had written "N/A" were excluded. Minimal alterations were made to autobiographical narratives, primarily addressing typos and repetitive words (e.g., the - hte; the the). As the alterations were minor, no distinct analysis was carried out on the

original narratives. We also clearly instructed the participants not to use the word of interest in their texts or descriptive word responses. No whole autobiographical narratives were removed.

**Quantifying the descriptive words and rating scales.** The descriptive words generated by the participants in Phase 1 and Phase 2 were analyzed by the BERT model (i.e., 'bert-base-uncased'[24]), where we used the embeddings in the last layer (i.e., layer 12). The BERT embeddings had 768 dimensions. The categorization was based on a vector consisting of; either the embeddings generated from the word data only (768 dimensions), the four rating scales only (4 dimensions), or a combination of the rating scales and the embeddings combined (4 + 768 dimensions). As the number of dimensions in the embeddings (768) was rather large relative to the size of the dataset, a data compression algorithm called Singular Value Decomposition was used to compress this vector so that the first dimensions contained the most important information about the original vector. To make the comparisons fair, or comparable, between the three analyses, this data compression algorithm was also applied to the rating scales-only analysis, although this was not computationally necessary as the number of dimensions was already low (i.e., 4). Data distribution was assumed to be normal but this was not formally tested.

**Categorizing the responses using ML.** Participant responses were categorized into one of the four emotions using multinomial logistic regression, where we ensured that testing and training were always conducted in different subsets of the data. The categorizations were evaluated by a 10-fold leave-out cross-validation procedure[39], where 90% of the data were used for training the multinomial model, and the generated model was applied to 10% left-out data points. This leave-out procedure was repeated ten times so that all data points received a predicted value. The groups in each fold were generated randomly, with the constraint that no narrative in the training dataset could be found in the test dataset. In each fold, the number of dimensions was optimized based on the training data set, by trying the first 1, 2, 3, 5, 7, 10, 14, 19, 26, 35, 46, 61, 80, 105, 137, 179, 234, 305, 397, 517, and 768 dimensions, where we selected the number of dimensions with the fewest number of miscategorizations. The optimized number of dimensions found in the training data set was then applied to the held-out test dataset. For the words-only analysis, the mean number of optimized dimensions over the ten folds was 70 with a standard deviation of 14. For the rating scale-only analysis, the mean number of dimensions used was 3.64 with a standard deviation of 0.48 (i.e., in most cases all dimensions were used).

To test hypothesis 3, we also conducted multiple linear regression to predict the continuous values for each of the four rating scales. These regression models were otherwise conducted and evaluated in the same way as the multinomial regression model described above.

**Word clouds.** The word clouds were generated by a semantic t-test[40] as this method allows comparison, without the parameter fittings as is required for logistic regression, for binary classifications, where we used binary discrimination between whether a word belonged to an emotional state compared to the other three states. The semantic t-test is a statistical method that calculates whether two sets of semantic representations differ from each other by calculating the semantic similarity scores. The first step was to summarize the semantic representation of the word responses generated from one emotion into one semantic representation and then normalize this vector to the length of one. The words from the other three emotions were summarized in the same way. A difference vector was then calculated by subtracting these two vectors from each other and normalizing the length of this vector to one. The semantic similarity between this difference vector and the vector describing each unique word was then calculated by taking the dot product between these two vectors (i.e., mathematically equivalent to the cosines of the angle between the vectors). A 10-fold leave-out procedure was implemented while calculating the difference vector so that this vector did not include

the to-be-measured word. A t-test was used to determine if the semantic similarity scores were significantly larger than zero, following Bonferroni correction for multiple comparisons. The same procedure was repeated for the four emotions to produce four distinct word clouds. The figure shows the 25 words with the highest t-values for each emotion, where all words were statistically significant.

### Reporting summary

Further information on research design is available in the Nature Portfolio Reporting Summary linked to this article.

## Results

### Predicted categorization

In line with our first hypothesis, the results showed that the percentage of correct categorization of emotional narratives was significantly higher when based on word responses (64%) compared to the total score of the four rating scales (44%) in Phase 2 for the non-professional group ($X^2$(1, 400) = 16.10, $p$ = 0.0001, $\varphi$ = 0.20 [0.10, 0.29])(see Fig. 2). Basing the categorization on individual items of the four rating scales (26 items in total, i.e., 9 items for PHQ-9, 7 for GAD-7, and 5 for SWLS and 5 HILS) yielded lower categorization accuracy (30%) ($X^2$(1, 400) = 8.41, $p$ = 0.0037, $\varphi$ = 0.14 [0.05, 0.24]).

As per our second hypothesis, there was no statistically significant difference in the accuracy of categorization between word responses only (63%) versus rating scales and word responses combined (64%) in Phase

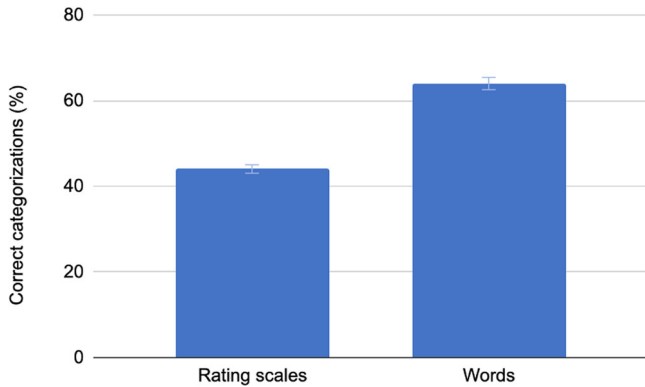

**Fig. 2 | Correct categorizations based on rating scales and descriptive word responses.** The error bars show the standard deviation of the mean ($N$ = 348).

2 for the non-professional group ($X^2$(1, 400) = 0.04, $p$ = 0.8355, $\varphi$ = 0.01 [−0.09, 0.11]). The accuracy of categorization was lower for rating scales compared to word responses in the cases of satisfaction ($X^2$(1, 100) = 5.88, $p$ = 0.0154, $\varphi$ = 0.25 [0.06, 0.42]) and anxiety ($X^2$(1, 100) = 45.63, $p$ < 0.01, $\varphi$ = 0.68 [0.55, 0.77]) (Table 2). The rating scale analysis consistently showed low accuracy for anxiety (i.e., 15% lower in all conditions), whereas the semantic analysis showed reasonably high accuracy for all emotional states (except for satisfaction in the smaller group of Phase 2 professionals). This suggests that the rating scale-based analysis was particularly poor at identifying anxiety.

Regarding our fourth, exploratory hypothesis, there was no statistically significant difference between the percentage of correct categorization of emotional states for word responses (56%) and rating scale responses (50%) in the healthcare professional group ($X^2$(1, 34) = 0.12, $p$ = 0.7260, $\varphi$ = 0.09 [−0.25, 0.41]). Table 2 presents a detailed breakdown and a comprehensive view of the results pertaining to the percentage of accurate categorizations for both rating scales and language measures across all data points.

### Accuracy and precision

The accuracy measure (i.e., the number of correct positive plus the number of correct negative categorizations divided by the total number of categorizations) is higher when the categorization is made on word responses compared to rating scales ($X^2$(1, 434) = 4.04, $p$ = 0.0445, $\varphi$ = 0.10 [0.00, 0.19]) (see Table 3). Similarly, the precision measure (i.e., the number of correct positive categorizations divided by the number of positive predictions) is also higher for categorizations made on word responses compared to rating scales ($X^2$(1, 434) = 20.88, $p$ < 0.001, $\varphi$ = 0.22 [0.13, 0.31]). Table 3 presents a breakdown of precision and accuracy values for each emotional state, providing a detailed depiction of the performance metrics specific to each category.

### Confusion and correlation matrices

Figure 3 shows a confusion matrix, or the number of times the model's prediction occurs with an emotional state in Phase 1. For the predictions based on rating scales (upper part of the table), most errors occurred when the model predicted depression, but the correct answer was anxiety ($N$ = 64). Based on words, the errors were more evenly spread out, and the highest number of errors occurred when the model predicted satisfaction, but the correct state was harmony ($N$ = 24).

Table 4 shows the Pearson correlation between the multinomial estimated coefficients. All absolute correlation values were larger for the rating scale model (0.68 < $r$ < 0.92) compared to the words-based model (0.07 < $r$ < 0.47), indicating a larger confusion for the rating scale model.

---

**Table 2 | Correct categorization divided into phase, model, and emotional state**

| Phase (-P/P) | Model | All (%) | Harmony (%) | Satisfaction (%) | Depression (%) | Anxiety (%) | N |
|---|---|---|---|---|---|---|---|
| 1 | RS | 39 | 46 | 21 | 65 | 15 | 297 |
| 1 | Words | 70 | 71 | 58 | 76 | 75 | 297 |
| 1 | Words + RS | 67 | 62 | 60 | 71 | 73 | 297 |
| 2(-P) | RS | 44 | 38 | 31 | 88 | 02 | 400 |
| 2(-P) | Words | 63 | 56 | 55 | 74 | 66 | 400 |
| 2(-P) | Words + RS | 64 | 57 | 58 | 75 | 65 | 400 |
| 2(P) | RS | 50 | 60 | 20 | 82 | 12 | 34 |
| 2(P) | Words | 56 | 70 | 20 | 45 | 75 | 34 |
| 2(P) | Words + RS | 59 | 70 | 20 | 55 | 75 | 34 |
| 1 & 2(-P) | RS | 42 | 43 | 27 | 78 | 08 | 731 |
| 1 & 2(-P) | Words | 66 | 63 | 55 | 73 | 70 | 731 |
| 1 & 2(-P) | Words + RS | 65 | 60 | 58 | 72 | 69 | 731 |

The table shows the percentage of correct categorizations (N(correct)/N (total)*100 divided into Phase 1 and Phase 2 for the non-professionals (-P) and professionals (P) and whether the predicted categorizations were based on the rating scales (RS) or word responses (Words).

## Inter-rater agreement

Table 5 presents the inter-rater agreement based on either rating scales or linguistic content, encompassing Phase 2 non-professionals' data as well as the combined data from non-professionals across both phases. The inter-rater agreement in categorization was not significantly greater for the rating scales than for the words in Phase 2 non-professionals' data ($X^2(1, 263) = 3.50$, $p = 0.062$, $\varphi = 0.12$ [−0.00, 0.24]) nor for the whole dataset ($X^2(1, 721) = 0.12$, $p = 0.724$, $\varphi = 0.01$ [−0.06, 0.09]).

The third hypothesis was evaluated by matching the narratives in Phases 1 and 2 and calculating the Pearson correlation between measures of HILS, SWILS, PHQ-9, and GAD-7. Here we compared the rating scale scores for these measures with corresponding descriptive word prediction of these rating scales. The descriptive word predictions were based on multiple-linear regression as described in the methods section. The results in Table 6 show that descriptive word responses have a significantly higher correlation (using Fisher's r to z transformation) than the corresponding rating scale measures, supporting Hypothesis 3.

## Word clouds of categorizations

Word clouds were produced to exhibit words with statistically significant t-values, elucidating which words best represented the four mental states (see Fig. 4). Words indicative of depression included sad and depressed, and words indicative of anxiety included anxious, worried, and nervous. Words indicative of harmony and satisfaction included "happy" and "content",

whereas "calm" was more central to harmony, and "hopeful" was more critical for satisfaction. Notice that the figure predominantly exhibits emotion-based words, where respondents largely employed such terms and the word cloud reflects this data-driven approach. However, symptoms can also be investigated with the QCLA approach, given that participants are prompted for this. See for example how somatic symptoms were investigated by Kjell and colleagues[30]. This study assessed the efficacy of how QCLA can capture items associated with both the primary and secondary symptoms linked to Major Depressive Disorder and Generalized Anxiety Disorder.

## Discussion

This study demonstrates that, on average, employing computational methods based on a single open-ended question with a five-word response format yielded higher accuracy in categorizing participant-generated narratives describing emotional states compared to using four standardized rating scales. This finding has significant implications as it indicates that open-ended descriptive word-based responses may have higher validity than the rating scales commonly used for mental health assessments. Furthermore, the effect size was statistically significant, where the percentage of correct categorizations for word responses overall was 64% compared to 44% for the rating scales, that is, the difference in the percentage of emotional states correctly categorized was 20%.

The results of this study are consistent with other studies showing that computational language assessment produces very strong correlations to the rating scales of harmony and satisfaction (e.g., $r = 0.84$) that rival the theoretical limits on test-retest reliability and inter-item correlations[30]. Extending beyond linguistic assessments, evidence from related fields, such as facial expressions and cooperative behavior, suggests that language responses may possess greater validity compared to traditional rating scales. For instance, Kjell and colleagues[29] explored the identification of facial expressions using word responses and rating scales, revealing a modest advantage for word responses (4%), though notably smaller than the advantage observed in our present study. Similarly, in another study[30] on cooperative behavior, word responses, in contrast to rating scales, predicted participants' cooperative behavior in a one-shot give-some dilemma game, where participants completed rating scales (HILS and SWLS) or word-response measures of harmony and satisfaction prior to conducting a GSDG (for details see Van Lange & Kuhlman[41]). Our argument for the superior categorization of word responses over rating scales in the present study stems from the independence of the outcome variable (grounded truth of the narratives) from the rating scales, a distinction from earlier studies on Quantitative Content Analysis (QCLA) that were often validated through correlations with rating scales[29]. This underscores the relevance of our current findings within the broader context of existing literature, emphasizing the advantages of employing computational methods with open-ended questions for assessing emotional states.

Some discrepancies were identified when categorization accuracy was subdivided between each separate emotional state. In Phase 2 (but not Phase 1), the rating scale classification of anxiety consistently demonstrated low accuracy precision. This implies a potential bias in the rating scale model, as it tends to assign low scores for anxiety, possibly at the expense of better capturing other emotions during the categorization process.

### Table 3 | Accuracy and precision measures

| Model | Measure | Harmony | Satisfaction | Depression | Anxiety |
|-------|---------|---------|--------------|------------|---------|
| RS | Accuracy | 76 | 66 | 68 | 79 |
| RS | Precision | 43 | 35 | 49 | 29 |
| Words | Accuracy | 82 | 79 | 80 | 86 |
| Words | Precision | 60 | 62 | 64 | 65 |

The table shows accuracy and precision measures, as percentages, for rating scales (RS) and word responses (Words) for harmony, satisfaction, depression, and anxiety for Phase 2 ($N = 434$) as validated from Phase 1.

|  | Rating Scales | | | | Words | | | |
|--|------|--------|------|------|------|--------|------|------|
|  | Har. | Satis. | Dep. | Anx. | Har. | Satis. | Dep. | Anx. |
| Harmony | 33 | 41 | 2 | 1 | 48 | 24 | 7 | 1 |
| Satisfaction | 38 | 34 | 8 | 16 | 25 | 60 | 7 | 4 |
| Depression | 15 | 34 | 108 | 64 | 7 | 20 | 90 | 23 |
| Anxiety | 0 | 0 | 5 | 2 | 6 | 5 | 18 | 55 |

**Fig. 3 | Confusion matrix.** The heat map shows the number of empirical emotional states (in rows) that co-occur with the number of predictions of emotional states for the multinomial models (in columns) in Phase 2 ($N = 369$). Dark green areas signify the highest co-occurrences, while dark red areas indicate the lowest. The following abbreviations were used: harmony (Har), satisfaction (Satis), depression (Dep), and anxiety (Anx).

### Table 4 | Correlation matrix

|  | Rating Scales | | | Words | | |
|--|------|--------|------|------|--------|------|
|  | Har. | Satis. | Dep. | Har. | Satis. | Dep. |
| Satis | 0.87 [0.85, 0.88] | | | −0.07 [−0.14, 0.01] | | |
| Dep | −0.92 [−0.93, −0.91] | −0.95 [−0.96, −0.95] | | −0.47 [−0.53, −0.41] | −0.42 [−0.48, −0.36] | |
| Anx | −0.88 [−0.89, −0.86] | −0.75 [−0.78, −0.71] | 0.68 [0.64, 0.72] | −0.41 [−0.47, −0.35] | −0.42 [−0.47, −0.35] | −0.21 [−0.27, −0.13] |

The table shows Pearson correlation scores for the multinomial coefficient estimates in Phase 2 ($N = 434$). The square brackets indicate the 95% confidence intervals. The following abbreviations were used: harmony (Har), satisfaction (Satis), depression (Dep), and anxiety (Anx).

## Table 5 | Inter-rater agreement

| Phase | Model | Agreement | Correct | N |
|-------|-------|-----------|---------|---|
| 2(-P) | RS | 90 | 79 | 263 |
| 2(-P) | Words | 82 | 65 | 263 |
| 1 & 2 (-P) | RS | 83 | 69 | 721 |
| 1 & 2 (-P) | Words | 82 | 66 | 721 |

The table shows the percentage of agreement between all pairwise categorizations of the same narratives for different rater models and phases. The columns show the phase, model, agreement (i.e., the percentage of identical categorizations), correct (i.e., the percentage agreement of correct categorizations), and the number of pairwise comparisons. The following abbreviations were used: non-professionals (-P), and rating scales (RS).

## Table 6 | Correlation between Phase 1 and Phase 2 measures matched on narratives

| | Rating Scales | Words | p |
|---|---------------|-------|---|
| HILS | 0.19 [0.09, 0.28] | 0.76 [0.09, 0.28] | <0.001 |
| SWLS | 0.47 [0.39, 0.54] | 0.76 [0.39, 0.54] | <0.001 |
| PHQ-9 | 0.48 [0.41, 0.55] | 0.77 [0.41, 0.55] | <0.001 |
| GAD-7 | 0.17 [0.07, 0.26] | 0.55 [0.07, 0.26] | <0.001 |

The table shows the Pearson correlation between Phase 1 and Phase 2 measures of HILS, SWLS, PHQ-9, and GAD-7 matched by narrative identity (N = 434). The square brackets indicate the 95% confidence intervals. The second column shows these values for rating scales (RS) and the third column for descriptive word predictions of these ratings scales. The p-values represent whether the two correlations differ using Fisher's r-to-z transformation.

Additional information was found by looking at the confusion matrix, where the number of incorrect predictions was higher for the rating scales in Phase 2 compared to word responses. The Pearson correlation between estimated coefficients was considerably higher for the rating scale models compared to the word-based models, suggesting that word responses discriminate better between emotional states. This is consistent with previous studies showing that word plots discriminate better than rating scales between related concepts[30,32].

A subset of the Phase 2 data comprised individuals currently employed in healthcare-related professions. This subgroup was incorporated into the study due to the anticipation that they have a more profound knowledge of the definition and assessment of depression, anxiety, harmony, and satisfaction. Nonetheless, this group did not result in higher accuracy of categorizations using the rating scales, and their nominal values of correct categorizations of the rating scales were less than that for the word responses. Nominally, the data looked similar to the larger non-professional control participant data; however, the number of professional subjects was too small (N = 34) to make it feasible to draw any firm conclusions.

In addition to having greater validity compared to rating scales, computational methods for language-based assessment of mental health have several other advantages. First, language is a natural way that people communicate their mental states. People prefer to communicate mental health with language rather than rating scales because they find language to be more precise and elaborate, and they prefer using language during communication with clinicians, although rating scales are seen as easier and faster[34]. Second, open-ended language responses allow for an idiosyncratic description of the participant's mental health, thus providing an opportunity for person-centered health care. This is very different from rating scales that measure a fixed construct defined by the research and where patients cannot add their person-centered view[29]. Third, the proposed language measure is short and thus quick to administer. As it stands, it can be conducted in a brief conversation by asking individuals to provide five words that describe their emotional state. In contrast, completing a single rating scale would pose a challenge within the same timeframe.

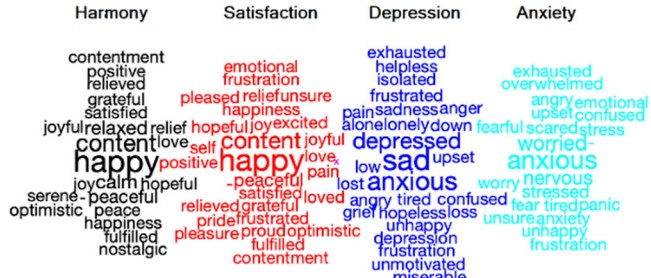

**Fig. 4 | Word clouds showing words indicative of the four emotional states.** The word clouds show from left to right harmony (black), satisfaction (red), depression (dark blue), and anxiety (light blue). The number of words in the dataset was N = 3787. The number of words in the figure is N = 100. The size of the words is proportional to the word frequency, and the most indicative words are in the center of the figure. Notice that some words (e.g., *happy*) were indicative of more than one emotion. Also, notice that words were generated by participants in both Phase 1 and 2, so the words that participants were not allowed to use in a specific condition in Phase 1 (e.g., "depression" in the depression condition), could be generated by Phase 2 participants, or by Phase 1 participants that were in another condition (i.e., "depression" was allowed to use in the "anxiety" condition in Phase 1). The image was created using SemanticExcel[38].

Our study suggests the potential benefits of using semantic measures such as QCLA in settings where standardized rating scales are used. The use of words instead of numerical scales can provide a more person-centered approach, which can help patients feel more understood and less depersonalized. For example, unstructured clinical notes are rarely made available in structured electronic health records, and a response format where patients are allowed to answer health-related questions in their own words rather than in a one-dimensional closed format presents numerous opportunities. The proposed method could facilitate diagnostic accuracy and treatment planning, ultimately improving treatment outcomes[31,42]. Unlike traditional rating scale methods, open-ended questions might be less likely to impose socially desirable, acquiescent responses or suggest likely symptoms e.g., "Are you having trouble relaxing?" (GAD-7)[4] and they are arguably a more natural form of expression[30]. Emotional support tools available digitally are proliferating and the academic community has recently observed a rise in social media text-mining studies[43,44]. The NLP method allows for the efficient evaluation of hundreds of predictors simultaneously and suggests economically sensitive solutions that can anticipate future outcomes, such as suicide actualization, attempts, or ideation[13]. Social media text mining, where written autobiographical accounts of one's state of mind are the primary means of communication, offers an alternative for preventative screening and detection of mental illness in the population, particularly in the prodromal phase, and for the assessment of risks for different mental health issues as a whole[17].

Healthcare-related data are well-positioned to provide insights into the health of our communities. However, one of the principal short-comings of the use of NLP in text analysis is the perceived privacy and ethics around scanning entire populations for mental health purposes. While it is true that NLP algorithms have the potential to analyze vast volumes of text data and detect patterns of mental health issues, doing so without appropriate ethical considerations could be seen as invasive and could likely lead to negative consequences for individuals and society as a whole. To ensure ethical use of such algorithms, it is important to have strict guidelines and regulations in place. For instance, any program or initiative using NLP to scan text for mental health should have clear protocols around data collection, sharing, storage, and usage to protect the privacy of individuals. The implementation of text analysis in a clinical context would therefore have the means to ensure respect for privacy and ethical considerations. With the introduction of the General Data Protection Regulation and efforts to mitigate social biases such as

race, ethnicity, and religion to make word embeddings more neutral[45–47], a balance between privacy and impartial intellectual advancement within clinical settings is underway. For that reason, we believe that QCLA has significant potential as a tool to be incorporated into the clinical setting.

## Limitations

The terms "anxiety" and "depression" are used differently in a clinical setting than by a layperson. Clinicians use the DSM-V definitions to assess mental health disorders, whereas participants writing about an event of depression or anxiety may use a broader understanding of the term that may or may not fit a clinician's assessment. Thus, the fairness of contrasting predictions obtained from clinical questionnaires with descriptive responses is a matter open to discussion. Nevertheless, consistent with the present study, individuals undergoing diagnostic assessments for clinical depression often express themselves in their own words. Comparing our sample with participants diagnosed with clinical depression could, in future research, offer insights and potentially address this matter in detail. The choice of a semantic approach or rating scale for the assessment of mental health depends on various factors, including the context and purpose of the assessment. If the goal is to acquire a simple standardized numerical evaluation of specific mental health aspects that can be compared with the previous literature, closed-ended rating scales may be suitable. However, open-ended questions may also be standardized to compare individuals from different samples. When aiming to encompass a broader range of emotions and experiences, particularly when respondents interpret emotions differently, open-ended responses can offer valuable qualitative perspectives. The intent is not to substitute one method for another but rather to complement traditional scales with semantic measures hence, for example, assisting primary diagnosis.

To further scrutinize the contrast between the differences between how individuals personally perceive the significance of conditions like depression and anxiety and what clinicians find relevant, it is important to capitalize on the distinction between the subjective feelings the individuals have and the more comprehensive, objective assessment required in a clinical setting. If NLP were to be applied in clinical practice, it might be beneficial to consider tailoring the model to the specific purposes of clinical evaluation, for instance, preventative screening, diagnosis, or following up on treatment progress. The present study did not stratify NLP predictions by gender, nationality, or education to investigate possible variations in the resulting model. Recognizing the need for customization to address the diverse dimensions of demographic factors and mental health states, including cognition, physiology, and behavior, could enhance the relevance and effectiveness of the assessment tool in real-world applications. The perpetuation of misdiagnoses and disparities observed with rating scales, particularly evident in situations like depression or schizophrenia diagnoses among black patients, must not be allowed to transfer to open-ended descriptive word responses analyzed using NLP and therefore requires thorough attention.

Another concern is that participants are instructed to report on past emotional episodes so that the emotional experience they had at the time might differ from the way they feel about it when writing the narrative. This concern would likely become particularly notable if individuals were to repeatedly recall the same experience, making them vulnerable to memory distortions and alterations during the memory reconsolidation process. Accurately recalling and describing emotional experiences may become challenging for individuals as it is reasonable to assume that individuals might have had different moods at different points in time and are situated in diverse contexts, potentially introducing data quality issues. In Phase 1 an independent set of participants generated narratives of self-experienced events relating to one of the four emotions; they were later read by participants in Phase 2, described in five words, and interpreted using rating scales commonly used for measuring the corresponding emotions. Thus, the success or failure of the categorization of the Phase 2 data depended on how participants interpreted the Phase 1 data, and participants in Phase 1 may have had different views of how these emotions had to be understood, or

interpreted, compared to how the rating scales are generally constructed. Although we find this possibility less likely because depression, anxiety, satisfaction, and harmony are commonly used concepts, relying solely on self-report measures of emotions may not provide a complete or accurate picture of emotional experiences.

We acknowledge that relying on past memory introduces potential challenges, such as recall bias, but it aligns with our goal of comparing the predictive capabilities of descriptive word responses with corresponding rating scales. Asking participants to share recent life episodes and using questionnaires based on their current feelings would be another valid approach. However, our specific research design aimed to establish a ground truth dataset, referring to the past recollection of emotional states in an autobiographical context. It prioritized the establishment of a historical baseline for a more comprehensive analysis of emotional experiences over time. Nonetheless, this aspect presents an intriguing avenue for prospective research, where the manipulation of mood could be explored, and a specialized model could be trained to identify and categorize specific emotions, among other factors. Also, a within-subjects study could be another possibility to account for repeated measures.

Finally, participants were constrained to offering only single-word responses, a method validated in previous research[29]. Importantly, this could result in an incomplete comprehension of pertinent symptoms. For instance, when individuals are prompted to articulate their emotional states during depressive episodes using descriptive words, there is a risk of overlooking relevant information. The use of descriptive has the potential to constrain the variety of states or symptoms being expressed with some necessitating more than a single word for an accurate depiction, as exemplified by phrases such as "eating less," "unable to concentrate," or "thinking about hurting myself." Considering the adaptable nature of the BERT model, there may be potential for reevaluating this constraint, where this model may not necessarily be bound by this limitation. BERT demonstrates flexibility by comprehending both phrases and even accommodating non-words and misspellings. Exploring the effectiveness of comparing single words to short phrases could be of interest to future research, as limited research has been done to compare these two approaches.

## Future work

The primary emphasis of the present study has been on the categorization of emotions. However, from a clinical standpoint, a research question that may hold additional interest is the exploration of the feasibility of applying a comparable method to classify narratives that detail participants' mental health into specific mental health diagnoses. Undertaking such a study carries direct implications for the assessment and diagnosis of mental health conditions, offering the potential for a more nuanced and comprehensive understanding of individuals' psychological well-being. This avenue of research could contribute valuable insights to the development of effective diagnostic tools and interventions in the realm of mental health.

The selection of BERT as the language model for our study was based on its widespread use and established standardization in NLP. Additionally, the choice was influenced by alignment with other research efforts within the research group[31]. While other language models, such as MentalBERT, could be reasonable alternatives, the decision to employ BERT in the present study was rooted in its proven performance and versatility across various NLP tasks. Future research could benefit from exploring different language models to enhance the comprehensiveness of our analyses.

## Conclusion

QCLA holds substantial promise as a tool to enhance the identification, possibly leading to improved treatment of various mental health disorders. Subsequent research involving larger and more diverse populations has the potential to delve deeper into the efficacy of QCLA and its broader applicability. The present study suggests that descriptive word responses offer an opportunity for a more precise and naturalistic assessment of psychological constructs and our findings show unprecedented accuracy in

the categorization of narratives on emotional states compared to one-dimensional numeric rating scales. This has important implications within the field of mental health because standardized rating scales, not open-ended language responses, are commonly used in quantitative assessment. Semantic measures may therefore constitute a cornerstone approach to a complementary assessment method of emotional states. Our work suggests that directed questions with responses to open-ended questions can be used to assess psychological constructs with higher validity and reliability than standardized numeric rating scales when analyzed by computational methods using NLP.

## Data availability

The original Qualtrics surveys, the description of the exact phrasing of the standardized rating scales, and anonymized participant data have been made publicly available at https://osf.io/gdkcb.

## Code availability

The corresponding author can provide the custom-made MATLAB computer code, upon request, in a manner that enables readers to replicate the published results. No code was used for preprocessing the data. The analysis made can be reproduced in semanticexcel.com without codes using the data in https://osf.io/gdkcb. The data can be copy pasted in the software.

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

## Acknowledgements

This work was supported by the Swedish Research Council Grant VR 2021–04627. The funders had no role in study design, data collection, analysis, decision to publish, or preparation of the manuscript. We thank all participants who took part in this study.

## Author contributions

S.S.: conceptualization, methodology, software, data interpretation, formal analysis, writing—review & editing, project administration, supervision, and funding acquisition. I.V.: conceptualization, methodology, investigation, data interpretation, formal analysis, writing—original draft preparation, review & editing, visualization. I.K.: methodology, investigation, writing—original draft preparation. N.E.: methodology, investigation, writing—original draft preparation.

## Funding

## Competing interests

The first author declares the following competing interests: S.S. is a co-founder and shareholder of a start-up that uses computational language assessments to diagnose mental health problems called Ablemind. The other authors declare no competing interests. There are no links between any of the companies mentioned in this text (e.g., Amazon or Alexa) and the authors or Ablemind.
