## [Peer Review File · Communications Psychology]

1st Nov 23

Dear Professor Sikström,

Thank you for your patience during the peer-review process--we apologize for the prolonged wait. Your manuscript titled "Question-Based Computational Language Approach Outperforms Rating Scales in Quantifying Emotional States" has now been seen by 3 reviewers, and I include their comments at the end of this message. They find your work of interest, but raised some important points. We are interested in the possibility of publishing your study in *Communications Psychology*, but would like to consider your responses to these concerns and assess a revised manuscript before we make a final decision on publication.

We therefore invite you to revise and resubmit your manuscript, along with a point-by-point response to the reviewers. Please highlight all changes in the manuscript text file.

Editorially, we consider it important that you address the requests for additional analyses, including a sensitivity analysis based on pre-processing exclusions, in your revision. Please address the reviewers' requests for additional details and clarifications regarding your methods, analyses, and models. Furthermore, please clearly indicate which hypotheses were preregistered and any deviations taken from the preregistration. All analyses proposed in the preregistration should be included in the manuscript or supplement.

Please use appropriate language to describe the results. There is no statistical test that can demonstrate absence of an effect. Statements such as 'There is no difference between x and y.' or 'X does not affect Y.' must be revised to read 'We found no statistically significant evidence of a difference between x and y.' or 'We found no statistically significant evidence that X affects Y.').

Please note that your revised manuscript must comply with our formatting and reporting requirements, which are summarized on the following checklist: Communications Psychology formatting checklist and also in our style and formatting guide Communications Psychology formatting guide.

Please use the following link to submit your revised manuscript, point-by-point response to the referees' comments (which should be in a separate document to any cover letter) and the completed checklist:

[link redacted]

We hope to receive your revised paper within 8 weeks; please let us know if you aren't able to

submit it within this time so that we can discuss how best to proceed. If we don't hear from you, and the revision process takes significantly longer, we may close your file. In this event, we will still be happy to reconsider your paper at a later date, provided it still presents a significant contribution to the literature at that stage.

Please do not hesitate to contact me if you have any questions or would like to discuss these revisions further. We look forward to seeing the revised manuscript and thank you for the opportunity to review your work.

Best regards,

Jennifer Bellingtier

Jennifer A. Bellingtier, PhD
Senior Editor
Communications Psychology

EDITORIAL POLICIES AND FORMATTING

Editorial Policy: Policy requirements (Download the link to your computer as a PDF.)

* **CODE AVAILABILITY:** All Communications Psychology manuscripts must include a section titled "Code Availability" at the end of the methods section. In the event of publication, we require that the custom analysis code supporting your conclusions is made available in a publicly accessible repository; at publication, we ask you to choose a repository that provides a DOI for the code; the link to the repository and the DOI will need to be included in the Code Availability statement. Publication as Supplementary Information will not suffice. We ask you to prepare code at this stage, to avoid delays later on in the process.

* **DATA AVAILABILITY:**

All Communications Psychology manuscripts must include a section titled "Data Availability" at the end of the Methods section or main text (if no Methods). More information on this policy, is

available at <http://www.nature.com/authors/policies/data/data-availability-statements-data-citations.pdf>.

At a minimum the Data availability statement must explain how the data can be obtained and whether there are any restrictions on data sharing. Communications Psychology strongly endorses open sharing of data. If you do make your data openly available, please include in the statement:

We recommend submitting the data to discipline-specific, community-recognized repositories, where possible and a list of recommended repositories is provided at <http://www.nature.com/sdata/policies/repositories>.

If a community resource is unavailable, data can be submitted to generalist repositories such as figshare or Dryad Digital Repository. Please provide a unique identifier for the data (for example a DOI or a permanent URL) in the data availability statement, if possible. If the repository does not provide identifiers, we encourage authors to supply the search terms that will return the data. For data that have been obtained from publicly available sources, please provide a URL and the specific data product name in the data availability statement. Data with a DOI should be further cited in the methods reference section.

REVIEWERS' EXPERTISE:

Reviewer #1 Natural Language Processing
Reviewer #2 Mental health, commutational models
Reviewer #3 Cognitive Modeling, NLP

REVIEWERS' COMMENTS:

Reviewer #1 (Remarks to the Author):

The authors advocate replacing close-ended rating scales with open-ended language responses for diagnosing mental health disorders. For this purpose, the authors test the performance of open-ended language responses over close-ended rating scales for predicting emotional states (depression, anxiety, satisfaction, and harmony). The participants were asked to generate narratives given four emotions namely depression, anxiety, satisfaction, and harmony. In parallel to narratives, the participants also filled out the questionnaires (PHQ-9, GAD-7, SWLS, and HILS) corresponding to these four emotion prompts. The narratives are then summarized into five descriptive words by two sets of participants. The task is to predict the emotion using (a) 5 descriptive words and (b) a close-ended rating scale score. The ground truth is the initial emotion prompt. The authors argue that the predictive performance of the model using five descriptive words is better than the one using a close-ended rating scale score.

This work is important as it takes into consideration the diverse manifestations of mental illness that can be captured using open-ended language responses. However, there are several unanswered questions:

Questions:

A Each question in the close-ended rating scales (e.g. PHQ-9) measures a verified aspect of mental health. It is also mentioned in the limitations that the participants may interpret the emotion terms in a colloquial sense or may not be depressed or suffering from anxiety in the clinical sense. In this case, closed-ended rating scale may have a low score but the open-ended response is discussing sadness or loneliness. How fair is it to compare the predictions obtained from clinical questionnaires vs with a descriptive response?

B It is not clear how these predictions can be used. For instance, the PHQ-9 score indicates the severity of depression. Here, how is the prediction of emotion state, “depression” or “harmony” helpful for clinicians?

C Line 267 – One of the major motivations behind open-ended language response is to capture diverse manifestations such as somatic symptoms (Line 94). However, there are only emotion-based words in Fig 3.

D It seems that the authors have only collected positive instances. E.g. describe an instance when you felt depression or anxiety. How were negative cases picked from open-ended responses for training ML models?

E Line 285: The close-ended rating scales filled based on past memory of the event. – Instead of asking participants to recall past experiences, would it not be better to ask participants to share recent life episodes and clinical questionnaires based on their current feelings to avoid errors?

Reviewer #2 (Remarks to the Author):

Thank you for the opportunity to review this interesting paper and study! Indeed, AI/NLP advances make it possible to transform the way we assess emotions and mental states, with free text potentially offering a more natural and human-centered approach. Accordingly, I think this paper has potential to contribute to the literature on mental health or mental state measurement. In particular, the authors make useful arguments about the limitations of using rating scales for evaluation (as not being human-centered or attuned to unique experience). They offer an important methodological contribution by pointing out limitations of prior approaches to validating free text assessment via comparison with rating scales (i.e., the assumption that rating scales are a gold-standard). They attempt to address this limitation by evaluating the validity of both approaches, using an independent outcome criteria. They also provide their study materials and data online (OSF), which facilitates open science and reproducibility. However, I have some questions and concerns about their interpretations and methodology, which I believe should be addressed and do not make the paper suitable for publication in its current state (please see below).

Introduction

- I would recommend elaborating more on why these specific constructs of anxiety, depression, satisfaction, and harmony were chosen. Also, why are these being described as emotional states?

This can lead to some confusion as to whether the authors are interested in measuring basic emotional states (i.e., sadness, happiness, fear, anger, etc)

- I'm not sure I agree with the point in lines 114-115 (i.e., "recent advancements in natural language processing (NLP) offer a potential solution for efficiently interpreting and quantifying language responses, while maintaining objectivity in the assessment process") – the application of NLP/analysis of free text can also be subjective/biased, especially with the application of LLMs, like BERT, which have been shown to express diagnostic and social stereotypes/biases (see for example, Zhang, H., Lu, A. X., Abdalla, M., McDermott, M., & Ghassemi, M. (2020, April). Hurtful words: quantifying biases in clinical contextual word embeddings. In proceedings of the ACM Conference on Health, Inference, and Learning (pp. 110-120) – but there are many other examples of this in the literature)

Methods

-the methods should contain a bit more in-depth description of the rating scales, the types of items included, how many and how they are scored

-how long did participants get to write the narratives? (can this be reported, alongside some summary statistics about the narratives, like word counts, etc?) What happens if participants did not write at least 100 words? Were they excluded?

-were participants allowed to write phrases as keywords? Or were they forced to use single words?

-there needs to be more information on the pre-processing step described in lines 334-335 – what was the specific exclusion criteria for removal of narratives and keywords? Can the authors briefly describe the types of narratives or words that were excluded? What was the criteria for removal due to repetition? (line 337) For example, if narratives and/or keywords unrelated to one of the four categorizations were removed (i.e., depression, anxiety, satisfaction, harmony), these steps could be introducing an advantage for the free-text vs rating scale approach. If so, then a sensitivity analysis not including manual preprocessing should be done and reported (i.e., including the narratives and keywords that were excluded/removed during this step) to determine whether results depend on these removals/exclusions? And in the discussion, the authors claim that a free-text approach can be adopted in a quick and easy way for measuring mental states (like depression), but is it feasible for these sorts of manual pre-processing steps to be carried out in research or clinical settings? This should be discussed.

-did the authors experiment with other LLMs? (for example, it seems that MentalBERT would be a reasonable LLM to use in this regard) - perhaps the choice of LLM could be justified

-can the authors clarify why summed scores were used for the rating scales and not BERT embeddings of highly endorsed items? This might provide an informative (and potentially fairer) comparison. Relatedly, lines 369-371 indicate that the authors also based categorization of rating scales on all their individual items vs. just summed scores, but this seems very unclear. Why were all items included and not just the ones that participants endorsed or reported high ratings for? For example, if a participant indicated 0 for the second item on the PHQ-9 when evaluating a time they felt harmonious or satisfied, was this item included as a predictor? Also, why is this analysis not described in the methods? And where are the specific results of this analysis provided? (I hope I am

understanding correctly what was done, but if not, please let me know!)

-could the authors clarify whether SVD was used to compress each of the three vectors of interest (ie word data, rating scales, word data + rating scales)? If so, could the authors clarify whether/how the vector for the rating scales was optimized, as there are only 4 dimensions in this vector.

-could the authors clarify whether the number of dimensions was optimized with 10-fold CV on the train set only and then the final optimal number for each vector of interest was evaluated in a hold-out test set? Currently, it looks like both optimization and evaluation were done on the same dataset.

-it would be useful to include percentages in Table 1 as well

Results

-the statement in lines 366-367 is unclear (“The categorization was significantly worse for the rating scales than the word response for depression, satisfaction, and harmony but was significantly better for anxiety”) – are the authors saying that categorization based on rating skills was better for anxiety? If so, where are these results presented? (This doesn’t appear to be the case in Table 2 but please let me know if I am missing something!)

-the authors should point to discrepancies between phase 1 and 2 data (especially for depression) – why might these have occurred? Perhaps confusion and correlation matrices broken down by phase could be helpful for interpreting these discrepancies as well

-the accuracy table is incorrectly referred to as table 3 (line 391)

-the results of the word clouds are a bit unclear and details related to their methodology should be presented in methods. But my main concern/question here is whether these were generated based on the keywords, and if so, why “depressed” and “anxious” appeared as keywords if phase 1 participants were asked not to use “depression” or “anxiety” as keywords?

-can the authors also clarify why the t-test/semantic similarity approach was used to generate word clouds, and why term frequency was not used (e.g., the top X keywords used to describe narratives about each construct/mental state)?

Discussion

-the authors’ overarching conclusion regarding more accurate categorization based on keywords vs rating scales is not entirely correct, given differences in findings between phases and constructs (which should be discussed)

-are there any potential limitations to the use of keyword generation as an assessment tool in clinical settings? Are there ways these might produce a limited understanding of relevant symptoms? For example, if we ask people to describe their emotional states related to depressive episodes using keywords, is it not possible that relevant information may be missed? (e.g., a participant may have changes in appetite or psychomotor agitation, but omit them from keywords) – also might use of keywords limit the types of states/symptoms being described? What if a state

requires more than one word to describe? (eg eating less, unable to concentrate, thinking about hurting myself) – I worry a bit that the discussion does not present a balanced view of both benefits and risks of keyword generation for assessment

-related to this, I agree that free text offers a more natural approach to assessment, but there is a broader issue of what is meaningful/salient to people experiencing a state or condition like depression and anxiety vs. what is meaningful/salient for clinical evaluation. For instance, while someone with depression may focus on feeling sad or unmotivated as the most meaningful/salient aspect of the state, a clinician may additionally consider changes in appetite and sleep as important indicators for assessment and care – so I'm not sure it's a simple matter of text providing a more accurate reflection of these mental states, but perhaps a different/person-centered reflection (and both can be valuable) – furthermore, participants were asked to generate keywords based on the "emotional state" whereas depression and anxiety involve other components (cognition, physiology, behaviour)

-the point about narrative writing being therapeutic is interesting, but perhaps not relevant here, as narratives were not used for prediction/categorization. Is there any evidence that writing keywords about emotional events or traumas is therapeutic? Also, the evidence for EW in depression or anxiety is still rather mixed (see for example Reinhold, M., Bürkner, P. C., & Holling, H. (2018). Effects of expressive writing on depressive symptoms—A meta-analysis. *Clinical Psychology: Science and Practice*, 25(1), e12224) (however it is better established for trauma, but trauma is not considered in this study)

-relatedly, the authors discuss the benefits of narrative writing and argue that QCLA can be a way to both assess and treat mental health, but then talk about how the production of keywords is quick and easy - narrative and keyword production are two very different processes with potentially different implications for both assessment and treatment?

-I appreciate the authors' discussions of privacy and ethics, but bias in LLMs and NLP methods should also be discussed (see my point above about biases related to social/racial stereotypes in BERT models, but these models also contain diagnostic stereotypes)

-also, the authors should describe some future directions for this work or how limitations of the current study might be addressed in future research

-finally, there are just some writing/clarity issues throughout the manuscript, see some examples below:

-Abstract: "We demonstrate that specific open-ended question analyzed by natural language processing (NLP) shows higher accuracy in categorizing emotional states compared to traditional rating scales" is a bit unclear as it wasn't an open ended question analyzed by NLP but participants were asked to generate 5 words and not an open ended response

-line 119- has AI/ML impacted the prediction of data? Or rather, it has advanced prediction based on data?

-line 121 – what does it mean that "Data analyzed using ML improves in accuracy as more data become available, and where the system learns a set of tasks without external intervention or supervision" – how does data improve in its accuracy? Or do you mean that the ML improves in accuracy with more data? And are the authors claiming that ML accuracy improves without supervision?

-inconsistent use of abbreviations (i.e., ML, machine learning)

-line 175: "text is generated by direct questions from participants" – the descriptions of QCLA studies are a bit unclear – do participants generate questions which are then used to assess mental health states?

-lines 192, 195 – I'm not sure that "language based methods/measures" is the best way to refer to this, since rating scales rely on language to some extent as well. A more specific descriptor would help (e.g., NLP on text responses or generated keywords)

Reviewer #3 (Remarks to the Author):

The manuscript titled "Question-Based Computational Language Approach Outperforms Rating Scales in Quantifying Emotional States" delves into the comparative efficacy of natural language processing (NLP) and traditional rating scales in quantifying emotional states. The abstract concisely outlines the study, highlighting that NLP, applied to specific open-ended questions, showcased a higher accuracy in categorizing emotional states compared to traditional rating scales. The study's methodology involves two groups of participants, with one group (N = 297) generating narratives related to four emotions (depression, anxiety, satisfaction, or harmony), and the second group (N = 434) reading these narratives. Both groups summarised these narratives using five descriptive words, in addition to rating stories using Likert scales. Machine learning managed to identify emotional labels more accurately when using semantic information (words) compared to relying on Likert scales only.

The manuscript is well written and it provides a robust data analysis. Some important references about works using semantic information for affective computing and psychometrics are missing from the state-of-art review but this is understandable, given how quickly the field is progressing. I have a few technical comments and clarifications about the text that might help the authors polish their already strong manuscript. For these reasons, I recommend acceptance with minor reviews.

In the abstract, it is not clear whether the first group rated their own stories right after their writing them or if that was done after group 2 produced its ratings. This is better explained on Page 10 so an easy re-wording of the abstract would be enough.

The adoption of word tagging for retrieving emotional states has been recently introduced as a cognitive task by the group of Thomas Hills, in their Emotional Recall Task, where participants were asked to enlist 10 emotional states they felt in the last month. In Li et al. (2020) the authors showed the Emotional Recall Task correlates with PANAS and can provide insights into different mental search strategies for individuals reporting high positive/negative affect levels. In a subsequent work, Fatima et al. (2021), the authors create DASentimental, an AI based on the Emotional Recall Task which learns how to map psychometric measurements of anxiety, stress and depression to word sequences produced by users. DASentimental is based on mental search strategies following cognitive science and associative knowledge modelling. These two works should be discussed in the literature review as they could strengthen the point of this current manuscript: cognitive information can inform AIs better about human affect.

In the Participants section - did the authors perform any sort of recognition of human participants

against bots? Automatic bots can be relatively frequent on platforms like Prolific.

How were the target categories depression, serenity, etc. selected? An explanation about this should enrich the Emotional Autobiographical Memory section in the Methods.

How many N/As were featured in the final data?

I understand Figure 2 provides not much more information compared to the in-text descriptions of accuracy. It would be better to replace Figure 2 with a confusion matrix, i.e. a 4x4 heatmap showing correct classifications but also mistakes across the 4 emotional dimensions. This would complement Table 4, which rather contains numbers but results difficult to read, as that Table mixes counts and continuous values without a separation.

The explanation for the semantic t-test in word clouds is not clear. Please extend that part or move it to the Methods with an additional formula. The note is clearer but it should be moved to the Methods. It would be more informative to couple the word cloud with a table ranking words and their t values.

In the Results, the author mention performing a 10-fold cross validation but report only values for accuracy without uncertainty margins, e.g. 64% rather than 63.8 ± 0.9 %. Please add the standard errors from the 10 folds to the results and comment on them - was the multinomial regression consistent in producing the same accuracy values?

In the Discussion it would be interesting to mention what would happen if the same person repeated the task over multiple times. Might it be that mood and context alter the word classifications for the very same piece of narratives? Could the current AI pipeline account for alterations provided by current mood and environment?

Typos:

and where the system learns -> and the system learns

scales or languages measures -> scales or language measures

Please find our comments to reviewers feedback related to the manuscript "Question-Based Computational Language Approach Outperforms Rating Scales in Quantifying Emotional States" (COMMSPSYCHOL-23-0238A). Our comments are made in red.

Reviewer #1 (Remarks to the Author):

The authors advocate replacing close-ended rating scales with open-ended language responses for diagnosing mental health disorders. For this purpose, the authors test the performance of open-ended language responses over close-ended rating scales for predicting emotional states (depression, anxiety, satisfaction, and harmony). The participants were asked to generate narratives given four emotions namely depression, anxiety, satisfaction, and harmony. In parallel to narratives, the participants also filled out the questionnaires (PHQ-9, GAD-7, SWLS, and HILS) corresponding to these four emotion prompts. The narratives are then summarized into five descriptive words by two sets of participants. The task is to predict the emotion using (a) 5 descriptive words and (b) a close-ended rating scale score. The ground truth is the initial emotion prompt. The authors argue that the predictive performance of the model using five descriptive words is better than the one using a close-ended rating scale score.

This work is important as it takes into consideration the diverse manifestations of mental illness that can be captured using open-ended language responses. However, there are several unanswered questions:

Questions:

A Each question in the close-ended rating scales (e.g. PHQ-9) measures a verified aspect of mental health. It is also mentioned in the limitations that the participants may interpret the emotion terms in a colloquial sense or may not be depressed or suffering from anxiety in the clinical sense. In this case, closed-ended rating scale may have a low score but the open-ended response is discussing sadness or loneliness. How fair is it to compare the predictions obtained from clinical questionnaires vs with a descriptive response?

Our response: The below reasoning has now been added to the limitations section of the manuscript.

One could argue that the fairness of comparing predictions derived from clinical questionnaires against descriptive responses is subject to debate. However, similar to the context of this study, individuals undergoing diagnostic testing for clinical depression often articulate their responses in their own words, even in the absence of a clinical diagnosis at that moment. Nevertheless, it would be valuable to compare our sample with one consisting of participants diagnosed with clinical depression, as this comparative analysis could yield insights and potentially address this issue. The preference for a semantic approach also hinges on various factors, such as the context and purpose of the assessment. If the objective is to obtain a standardized, numerical evaluation of specific mental health aspects, closed-ended rating scales might be more suitable. Conversely, if the aim is to encompass a broader range of emotions and experiences, open-ended responses can provide valuable qualitative perspectives.

B It is not clear how these predictions can be used. For instance, the PHQ-9 score indicates the severity of depression. Here, how is the prediction of emotion state, “depression” or “harmony” helpful for clinicians?

Our response: The intention is not to replace one method with another but rather to supplement traditionally used scales with semantic measures, hence, for example, assisting primary diagnosis.

This point has been added to the discussion section of the manuscript.

C Line 267 – One of the major motivations behind open-ended language response is to capture diverse manifestations such as somatic symptoms (Line 94). However, there are only emotion-based words in Fig 3.

Our response: To deal with this comment, we have added the following Notice that the figure predominantly exhibits emotion-based words, where respondents largely employed such terms and the word cloud is reflecting this data-driven approach. However, symptoms can also be investigated with QCLA approach, given that participants are prompted for this. See for example how somatic symptoms were investigated in Kjell et al.'s (2021). This study assessed the efficacy of how QCLA can capture items associated with both the primary and secondary symptoms linked to Major Depressive Disorder and Generalized Anxiety Disorder.

D It seems that the authors have only collected positive instances. E.g. describe an instance when you felt depression or anxiety. How were negative cases picked from open-ended responses for training ML models?

Our response: We did not collect instances specifically focused on not having depression or anxiety. We believe that e.g., low harmony or satisfaction would correlate with high depression and anxiety and vice versa. In our dataset, we encouraged participants to share their experiences across different emotional states. To ensure a balanced representation for training machine learning models, we sampled participants randomly, allowing for a large range of emotional responses from the participants. This approach allowed us to create a dataset that reflects the complexity and diversity of human emotions, fostering a more comprehensive understanding of the development and training of our models.

E Line 285: The close-ended rating scales filled based on past memory of the event. – Instead of asking participants to recall past experiences, would it not be better to ask participants to share recent life episodes and clinical questionnaires based on their current feelings to avoid errors?

Our response: The following point has been added to the manuscript in response to this reviewer's comment: While asking participants to share recent life episodes and using clinical questionnaires based on their current feelings is a valid approach, our specific research design aimed to establish a ground truth dataset. In our case, the term "ground truth" refers to the past recollection of emotional states in an autobiographical context. This approach allows us to gather data that goes beyond participants' current feelings and taps into the richness of their historical experiences. We acknowledge that relying on past memory introduces potential challenges, such as recall bias, but it aligns with our goal of understanding the long-term impact of emotional experiences.

Moreover, employing closed-ended rating scales filled based on past memory provides a standardized method for participants to articulate their emotions, aiding in the creation of a structured dataset for training and validation. While there are advantages to collecting real-time data on current feelings, our methodology prioritizes the establishment of a historical baseline for a more comprehensive analysis of emotional experiences over time. Each approach has its merits, and our choice was guided by the specific objectives of our study.

Reviewer #2 (Remarks to the Author):

Thank you for the opportunity to review this interesting paper and study! Indeed, AI/NLP advances make it possible to transform the way we assess emotions and mental states, with free text potentially offering a more natural and human-centered approach. Accordingly, I think this paper has

potential to contribute to the literature on mental health or mental state measurement. In particular, the authors make useful arguments about the limitations of using rating scales for evaluation (as not being human-centered or attuned to unique experience). They offer an important methodological contribution by pointing out limitations of prior approaches to validating free text assessment via comparison with rating scales (i.e., the assumption that rating scales are a gold-standard). They attempt to address this limitation by evaluating the validity of both approaches, using an independent outcome criteria. They also provide their study materials and data online (OSF), which facilitates open science and reproducibility. However, I have some questions and concerns about their interpretations and methodology, which I believe should to be addressed and do not make the paper suitable for publication in its current state (please see below).

Introduction

- I would recommend elaborating more on why these specific constructs of anxiety, depression, satisfaction, and harmony were chosen. Also, why are these being described as emotional states? This can lead to some confusion as to whether the authors are interested in measuring basic emotional states (i.e., sadness, happiness, fear, anger, etc).

Our response: We have added the following response to the introduction of the manuscript (p. 10). The present study compared language-based responses and rating scales using an outcome criterion that is independent of rating scales. In particular, we focused on an outcome criterion where participants were instructed to generate self-experienced narratives with specific emotional content related to key psychological constructs either depression, anxiety, satisfaction, or harmony (referred to as emotional states). These four psychological constructs represent a blend of positive psychology and clinical psychology elements, each characterized by a specific emotional valence that can be quantified through the use of standardized rating scales. The selection of these four psychological constructs aligns with previous works from our research group, ensuring consistency and facilitating the comparability of findings across studies.

- I'm not sure I agree with the point in lines 114-115 (i.e., "recent advancements in natural language processing (NLP) offer a potential solution for efficiently interpreting and quantifying language responses, while maintaining objectivity in the assessment process") – the application of NLP/analysis of free text can also be subjective/biased, especially with the application of LLMs, like BERT, which have been shown to express diagnostic and social stereotypes/biases (see for example, Zhang, H., Lu,

A. X., Abdalla, M., McDermott, M., & Ghassemi, M. (2020, April). Hurtful words: quantifying biases in clinical contextual word embeddings. In proceedings of the ACM Conference on Health, Inference, and Learning (pp. 110-120) – but there are many other examples of this in the literature)

Our response: We appreciate your perspective, and you raise a valid point. We may be oversimplifying the challenges associated with NLP applications, particularly with models like BERT. We have included the point in the manuscript (p.9), including the reference to the literature you provided as follows: While it is true that BERT comes with its own limitations, including subjectivity and biases that may reflect diagnostic and social stereotypes, it has undeniably made a significant breakthrough in the field of NLP. BERT's contribution lies in its introduction of unsupervised pre-trained models, allowing them to draw insights from extensive sets of unlabeled text data (Rogers et al., 2020). Recent work by Schwartz et al. (2022) reflects a commitment to advancing responsible and ethical practices in this rapidly developing field.

Methods

-the methods should contain a bit more in-depth description of the rating scales, the types of items included, how many and how they are scored

Our response: The in-depth description of the rating scales has now been added to the manuscript, measures section:

The Patient Health Questionnaire (PHQ-9) is a widely used tool to assess depression severity. It consists of nine questions that correspond to the criteria for diagnosing major depressive disorder in the DSM-IV. The nine items on the PHQ-9 cover various symptoms of depression, such as mood, sleep, appetite, and energy levels. Each item is scored on a scale from 0 to 3. Participants' responses to the nine items are then added together to obtain a total score, ranging from 0 to 27. The total score indicates the severity of depression: minimal, mild, moderate, moderately severe, and severe depression.

The Generalized Anxiety Disorder (GAD-7) scale is another commonly used tool in mental health assessments, specifically designed to measure the severity of generalized anxiety disorder. The seven items on the GAD-7 are scored on a scale from 0 to 3. Participants' responses to the seven items are then added together to obtain a total score, ranging from 0 to 21. The total score indicates the severity of generalized anxiety disorder: minimal, mild, moderate, and severe anxiety.

The Satisfaction With Life Scale (SWILS) is a widely used tool to measure an individual's overall life satisfaction. The Harmony in Life scale (HILS) measures a person's overall satisfaction and contentment with life. Both HILS and SWILS scales consist of five statements that participants respond to on a scale from 1 to 7. Participants' responses to these items are then summed up to obtain a total score, which can range from 5 to 35. Higher scores indicate higher levels of life satisfaction/harmony in life, while lower scores suggest lower satisfaction/harmony in life.

-how long did participants get to write the narratives? (can this be reported, alongside some summary statistics about the narratives, like word counts, etc?) What happens if participants did not write at least 100 words? Were they excluded?

Our response: Participants were given the freedom to write narratives for as long as they desired. Participants who did not meet the minimum word requirement were not allowed to proceed further in the survey, ensuring that all collected responses met a basic length threshold for analysis. Also, our wording was mistaken and there was no pre-specified number of words that participants had to enter. We have now corrected that. Participants were in fact required to write "at least a paragraph (approximately five sentences)". For the narratives, there was no minimum or maximum number of words. On average, participants wrote 81.32 words (SD = 40.66). The narratives underwent a manual review to guarantee the exclusion of any nonsensical data and no narratives had to be removed as a result.

The details have now been added to the relevant methods sections.

-were participants allowed to write phrases as keywords? Or were they forced to use single words?

*Our response and this consideration have been added in the discussion section (p.41) and yet again highlighted in the procedure section of the methods - single words **only** were used: Participants were constrained to offering only single-word responses, a method validated in previous research (Kjell et al., 2019). Importantly, this could result in an incomplete comprehension of pertinent symptoms. For instance, when individuals are prompted to articulate their emotional states during depressive episodes using keywords, there is a risk of overlooking relevant information. The use of descriptive has the potential to constrain the variety of states or symptoms being expressed with some necessitating more than a single word for an accurate depiction, as exemplified by phrases such as "eating less," "unable to concentrate," or "thinking about hurting myself." Considering the adaptable nature of the BERT model, there may be potential for reevaluating this constraint. With the use of the BERT model, we are not necessarily bound by this limitation. This model demonstrates flexibility by comprehending both phrases and even accommodating non-words and misspellings. Exploring the effectiveness of comparing single words to short phrases could be of interest to future research, as limited research has been done to compare these two approaches.*

-there needs to be more information on the pre-processing step described in lines 334-335 – what was the specific exclusion criteria for removal of narratives and keywords? Can the authors briefly describe the types of narratives or words that were excluded? What was the criteria for removal due to repetition? (line 337) For example, if narratives and/or keywords unrelated to one of the four categorizations were removed (i.e., depression, anxiety, satisfaction, harmony), these steps could be introducing an advantage for the free-text vs rating scale approach. If so, then a sensitivity

analysis not including manual preprocessing should be done and reported (i.e., including the narratives and keywords that were excluded/removed during this step) to determine whether results depend on these removals/exclusions? And in the discussion, the authors claim that a free-text approach can be adopted in a quick and easy way for measuring mental states (like depression), but is it feasible for these sorts of manual pre-processing steps to be carried out in research or clinical settings? This should be discussed.

Our response: These are all very valid points that have helped us shape our description of data pre-processing steps. These have now been added to the manuscript, statistical analysis section.

No whole narratives were removed as a result of this process. Minimal alterations were made to texts, primarily addressing typos and repetitive words (e.g., the - hte; the the). As the alterations were minor in nature, no distinct analysis was carried out on them. We also clearly instructed the participants not to use the word of interest in their texts or descriptive word responses.: "Please do not use the word [depression/anxiety/satisfaction/harmony] in your text." and so introducing no advantage for the free-text vs rating scale approach.

As for feasibility of carrying out these steps in research/clinical settings: the feasibility of manual pre-processing depends on factors such as the scale of the data, available resources, and the specific goals of the study or clinical analysis. For large datasets, automated pre-processing techniques may be more efficient, but in cases where precision and attention to detail are crucial, manual pre-processing steps may still be preferred. Manual pre-processing is a common practice in various fields, including research and clinical studies. BERT is able to disregard repetitive words as it can be tailored to suit specific clinical/research purposes. Future studies might explore the impact of manual versus automated pre-processing procedures on the classification of emotional states.

-did the authors experiment with other LLMs? (for example, it seems that MentalBERT would be a reasonable LLM to use in this regard) - perhaps the choice of LLM could be justified

Our response that has been added to the future work section of the manuscript: The selection of BERT as the language model for our study was based on its widespread use and established standardization in NLP. Additionally, the choice was influenced by alignment with other research efforts within our group. While we acknowledge that other language models, such as MentalBERT, could be reasonable alternatives, our decision to employ BERT was rooted in its proven performance and versatility across various NLP tasks. However, we appreciate the suggestion, and in future research, it could be beneficial to explore and justify the choice of different language models to enhance the comprehensiveness of our analyses.

-can the authors clarify why summed scores were used for the rating scales and not BERT embeddings of highly endorsed items? This might provide an informative (and potentially fairer) comparison. Relatedly, lines 369-371

indicate that the authors also based categorization of rating scales on all their individual items vs. just summed scores, but this seems very unclear. Why were all items included and not just the ones that participants endorsed or reported high ratings for? For example, if a participant indicated 0 for the second item on the PHQ-9 when evaluating a time they felt harmonious or satisfied, was this item included as a predictor? Also, why is this analysis not described in the methods? And where are the specific results of this analysis provided? (I hope I am understanding correctly what was done, but if not, please let me know!)

Our response: We removed the sentence on lines 369-371, as the result of the proposed analysis did not improve the accuracy, and because it may lead to confusion about what we analyzed. The concerns raised in that paragraph have been addressed and resolved.

Just to clarify the method and this information is now clarified in the methods section, for the rating scales categorization, we did a multinomial logistic regression on the total score of the four rating scales combined, so that the regression had four numerical values as input. For the BERT embeddings categorization, we used all BERT dimensions so that their multiple logistic regression coefficients had 768 inputs. For the combined rating scale and BERT categorization, we used the total score of the four rating scales and the BERT embeddings together so that their multiple logistic regression had 772 inputs.

-could the authors clarify whether SVD was used to compress each of the three vectors of interest (ie word data, rating scales, word data + rating scales)? If so, could the authors clarify whether/how the vector for the rating scales was optimized, as there are only 4 dimensions in this vector.

Our response: We have now clarified that the same data compression algorithm was used for all analyses to make results comparable: "As the number of dimensions in the embeddings (768) was rather large relative to the size of the dataset, a data compression algorithm called Singular Value Decomposition (SVD) was used to compress this vector so that the first dimensions contained the most important information about the original vector. To make the comparisons fair, or comparable, between the three analyses, this data compression algorithm was also applied to the rating scales-only analysis, although this was not computationally necessary as the number of dimensions already were low (i.e., 4)."

-could the authors clarify whether the number of dimensions was optimized with 10-fold CV on the train set only and then the final optimal number for each vector of interest was evaluated in a hold-out test set? Currently, it looks like both optimization and evaluation were done on the same dataset.

Our response: It is now clarified that the optimization of the number of dimensions was made in the training data set. The selected number of dimensions was then applied to the held-out test data set.

-it would be useful to include percentages in Table 1 as well

Our response: Done

Results

-the statement in lines 366-367 is unclear (“The categorization was significantly worse for the rating scales than the word response for depression, satisfaction, and harmony but was significantly better for anxiety”) – are the authors saying that categorization based on rating skills was better for anxiety? If so, where are these results presented? (This doesn’t appear to be the case in Table 2 but please let me know if I am missing something!)

Our response: Apologies for confusion. It was a mistake. We had written anxiety but it was meant to be depression. It has now been addressed. See it corrected in the second sentence of the results section.

-the authors should point to discrepancies between phase 1 and 2 data (especially for depression) – why might these have occurred? Perhaps confusion and correlation matrices broken down by phase could be helpful for interpreting these discrepancies as well

Our response: Following the correction in the writing, we now notice that the rating scales consistently make poor predictions for anxiety, whereas all results are reasonably good for the semantic measures (except satisfaction for the small group of professionals in Phase 2). This note has been noted in the results section (see the second paragraph).

The above considerations have been added to the relevant results and discussion sections.

-the accuracy table is incorrectly referred to as table 3 (line 391)

Our response: We believe that the current writing is correct as Table 2 is about categorization and Table 3 is about accuracy and precision.

-the results of the word clouds are a bit unclear and details related to their methodology should be presented in methods. But my main concern/question here is whether these were generated based on the keywords, and if so, why “depressed” and “anxious” appeared as keywords

if phase 1 participants were asked not to use “depression” or “anxiety” as keywords?

Our response: Figure 3 includes words generated by all participants in both Phase 1 and 2, so the words that participants were not allowed to use in a specific condition in Phase 1, could be generated by Phase 2 participants, or by Phase 1 participants that were in another condition (i.e., “depression” was allowed to use in the “anxiety” condition in Phase 1).

-can the authors also clarify why the t-test/semantic similarity approach was used to generate word clouds, and why term frequency was not used (e.g., the top X keywords used to describe narratives about each construct/mental state)?

Our response: The word clouds were generated by a semantic t-test (Kjell et al., 2020) as this method is suitable for binary classifications, and we used binary discrimination between whether a word belonged to an emotional state compared to the other three states.

Discussion

-the authors’ overarching conclusion regarding more accurate categorization based on keywords vs rating scales is not entirely correct, given differences in findings between phases and constructs (which should be discussed)

Our response: We have addressed the choice of words in this section. Indeed, on average, the overall accuracy of categorizing participant-generated narratives describing emotional states was higher when utilizing computational methods based on a single open-ended question with a response format of five words, as opposed to employing four standardized rating scales.

We discuss this shortcoming by recognizing that some discrepancies were identified when categorization accuracy was subdivided between each separate emotional state. In Phase 2 (but not Phase 1), the rating scale classification of depression consistently demonstrated greater precision than the semantic measure. This implies a potential bias in the model, as it tends to assign higher scores for depression, possibly at the expense of capturing other emotions during the categorization process. This bias might stem from the heightened salience of depression as a psychological concept. Participants in Phase 2 may be more confident in responding to the rating scale when it pertains to depression, possibly due to its familiarity and ease of addressing this concept compared to the other three psychological constructs examined in the study.

-are there any potential limitations to the use of keyword generation as an assessment tool in clinical settings? Are there ways these might produce a

limited understanding of relevant symptoms? For example, if we ask people to describe their emotional states related to depressive episodes using keywords, is it not possible that relevant information may be missed? (e.g., a participant may have changes in appetite or psychomotor agitation, but omit them from keywords) – also might use of keywords limit the types of states/symptoms being described? What if a state requires more than one word to describe? (eg eating less, unable to concentrate, thinking about hurting myself) – I worry a bit that the discussion does not present a balanced view of both benefits and risks of keyword generation for assessment

Our response: Certainly, participants were constrained to offering only single-word responses, a method validated in previous research (Kjell et al., 2019). However, with the use of the BERT model, we are not necessarily bound by this limitation. The BERT model demonstrates flexibility by comprehending both phrases and even accommodating non-words and misspellings. Considering the adaptable nature of the BERT model, there may be potential for reevaluating this constraint could be beneficial. Exploring the effectiveness of comparing single words to short phrases could be of interest in the future, as less research has compared these two approaches.

Please see the comment addressed in the manuscript, especially the limitation section which should now represent a more balanced view of both the benefits and limitations of the use of NLP for the assessment of psychological constructs.

-related to this, I agree that free text offers a more natural approach to assessment, but there is a broader issue of what is meaningful/salient to people experiencing a state or condition like depression and anxiety vs. what is meaningful/salient for clinical evaluation. For instance, while someone with depression may focus on feeling sad or unmotivated as the most meaningful/salient aspect of the state, a clinician may additionally consider changes in appetite and sleep as important indicators for assessment and care – so I'm not sure it's a simple matter of text providing a more accurate reflection of these mental states, but perhaps a different/person-centered reflection (and both can be valuable) – furthermore, participants were asked to generate keywords based on the “emotional state” whereas depression and anxiety involve other components (cognition, physiology, behaviour)

Our response: We agree, this is a valid point you raise regarding the distinction between what is meaningful for individuals experiencing depression and anxiety versus what holds significance in a clinical evaluation context. Certainly, there is a difference between the subjective experience of these emotional states and the comprehensive assessment required in a clinical setting. If this approach were to be applied in clinical practice, it might be beneficial to consider tailoring the models to the specific purposes of clinical evaluation, for instance, preventative screening,

diagnosis, or following up on treatment progress. Recognizing the need for customization to address the diverse dimensions of mental health states, including cognition, physiology, and behavior, could enhance the relevance and effectiveness of the assessment tool in real-world applications.

The above paragraph has now been added to the discussion.

-the point about narrative writing being therapeutic is interesting, but perhaps not relevant here, as narratives were not used for prediction/categorization. Is there any evidence that writing keywords about emotional events or traumas is therapeutic? Also, the evidence for EW in depression or anxiety is still rather mixed (see for example Reinhold, M., Bürkner, P. C., & Holling, H. (2018). Effects of expressive writing on depressive symptoms—A meta-analysis. *Clinical Psychology: Science and Practice*, 25(1), e12224) (however it is better established for trauma, but trauma is not considered in this study)

Our response: We agree with the reviewer that this point is not relevant, and we have therefore removed it.

-relatedly, the authors discuss the benefits of narrative writing and argue that QCLA can be a way to both assess and treat mental health, but then talk about how the production of keywords is quick and easy - narrative and keyword production are two very different processes with potentially different implications for both assessment and treatment?

Our response: Same here, this point has been removed.

-I appreciate the authors' discussions of privacy and ethics, but bias in LLMs and NLP methods should also be discussed (see my point above about biases related to social/racial stereotypes in BERT models, but these models also contain diagnostic stereotypes)

Our response: I appreciate you raising this point. The introduction and discussion sections now address the significance of mitigating bias in BERT models.

-also, the authors should describe some future directions for this work or how limitations of the current study might be addressed in future research

Our response: A section of future work has been added to the discussion
"Future work"

The primary emphasis of the present study has been on the categorization of emotions. However, from a clinical standpoint, a research question that may hold additional interest is the exploration of the feasibility of applying a comparable method to classify narratives that detail participants' mental health into specific mental health diagnoses. Undertaking such a study carries direct implications for the assessment and diagnosis of mental health conditions, offering the potential for a more nuanced and comprehensive understanding of individuals' psychological well-being. This avenue of research could contribute valuable insights to the development of effective diagnostic tools and interventions in the realm of mental health.

The selection of BERT as the language model for our study was based on its widespread use and established standardization in NLP. Additionally, the choice was influenced by alignment with other research efforts within our group. While we acknowledge that other language models, such as MentalBERT, could be reasonable alternatives, our decision to employ BERT was rooted in its proven performance and versatility across various NLP tasks. Future research could benefit from exploring different language models to enhance the comprehensiveness of our analyses."

-finally, there are just some writing/clarity issues throughout the manuscript, see some examples below:

-Abstract: "We demonstrate that specific open-ended question analyzed by natural language processing (NLP) shows higher accuracy in categorizing emotional states compared to traditional rating scales" is a bit unclear as it wasn't an open ended question analyzed by NLP but participants were asked to generate 5 words and not an open ended response

Our response: That was confusing wording. The issue has now been addressed. See the updated version of the abstract.

-line 119- has AI/ML impacted the prediction of data? Or rather, it has advanced prediction based on data?

Our response: The wording might indeed have been rather misleading, it has now been addressed. AI/ML has impacted the advancement of prediction based on data. They have improved the accuracy of predicting outcomes by analyzing large datasets and identifying patterns that may not be readily apparent through traditional methods such as manual analysis, or simpler statistical methods that do not encompass the complex learning capabilities, e.g., basic statistical modeling, and simple regression analyses. Therefore, AI/ML has not only influenced the prediction process itself but has also elevated the overall quality and precision of predictions through data-driven approaches.

-line 121 – what does it mean that "Data analyzed using ML improves in accuracy as more data become available, and where the system learns a set of tasks without external intervention or supervision" – how does data improve in its accuracy? Or do you mean that the ML improves in accuracy

with more data? And are the authors claiming that ML accuracy improves without supervision?

Our response: Yes, the writing was unclear. We have now clarified this part and indicated that the capability for a system to autonomously learn tasks without external intervention or supervision is indeed a characteristic often associated with ML, specifically in the context of unsupervised learning. Unsupervised learning algorithms can discover patterns and relationships within data without explicit guidance or labeled examples from humans. BERT, the model used in this study, is an NLP model that falls under the category of unsupervised learning. BERT is trained on large amounts of unlabeled text data, learning to understand the contextual relationships and meanings of words within sentences. Yet, the model can be fine-tuned on specific tasks using labeled data to make it more task-specific. For example, it can be fine-tuned for tasks like sentiment analysis, named entity recognition or question answering.

-inconsistent use of abbreviations (i.e., ML, machine learning)

Our response: Corrected

-line 175: “text is generated by direct questions from participants” – the descriptions of QCLA studies are a bit unclear – do participants generate questions which are then used to assess mental health states?

Our response: Yes, thank you for the remark. The sentence has now been adjusted as follows: Kjell and colleagues (2019) developed a Question-based Computational Language Assessment (QCLA) where text is generated by asking participants to answer open-ended questions from participants that can be transformed into a quantifiable vector using NLP.

-lines 192, 195 – I’m not sure that “language based methods/measures” is the best way to refer to this, since rating scales rely on language to some extent as well. A more specific descriptor would help (e.g., NLP on text responses or generated keywords)

Our response: Yes, thank you. More appropriate wording has now been used.

Reviewer #3 (Remarks to the Author):

The manuscript titled "Question-Based Computational Language Approach Outperforms Rating Scales in Quantifying Emotional States" delves into the comparative efficacy of natural language processing (NLP) and traditional rating scales in quantifying emotional states. The abstract concisely outlines the study, highlighting that NLP, applied to specific open-ended questions, showcased a higher accuracy in categorizing emotional states compared to traditional rating scales. The study's methodology involves two

groups of participants, with one group (N = 297) generating narratives related to four emotions (depression, anxiety, satisfaction, or harmony), and the second group (N = 434) reading these narratives. Both groups summarised these narratives using five descriptive words, in addition to rating stories using Likert scales. Machine learning managed to identify emotional labels more accurately when using semantic information (words) compared to relying on Likert scales only.

The manuscript is well written and it provides a robust data analysis. Some important references about works using semantic information for affective computing and psychometrics are missing from the state-of-art review but this is understandable, given how quickly the field is progressing. I have a few technical comments and clarifications about the text that might help the authors polish their already strong manuscript. For these reasons, I recommend acceptance with minor reviews.

In the abstract, it is not clear whether the first group rated their own stories right after their writing them or if that was done after group 2 produced its ratings. This is better explained on Page 10 so an easy re-wording of the abstract would be enough.

Our response: Thank you for the note, it has now been corrected.

The adoption of word tagging for retrieving emotional states has been recently introduced as a cognitive task by the group of Thomas Hills, in their Emotional Recall Task, where participants were asked to enlist 10 emotional states they felt in the last month. In Li et al. (2020) the authors showed the Emotional Recall Task correlates with PANAS and can provide insights into different mental search strategies for individuals reporting high positive/negative affect levels. In a subsequent work, Fatima et al. (2021), the authors create DASentimental, an AI based on the Emotional Recall Task which learns how to map psychometric measurements of anxiety, stress and depression to word sequences produced by users. DASentimental is based on mental search strategies following cognitive science and associative knowledge modelling. These two works should be discussed in the literature review as they could strengthen the point of this current manuscript: cognitive information can inform AIs better about human affect.

Our response: Thanks for your insightful observation. Indeed, the field of affective computing and psychometrics is evolving rapidly, and it is tough to encompass all the relevant references in a single review. For the wholesome overview of the literature in the introduction, in the introduction section, we added references to the work of Li and colleagues relating to the Emotional Recall Task as follows:

Recent studies have shown that text-based answers analyzed by computational methods can indeed predict corresponding close-ended rating scales such as PANAS, Ryff's Scales of Psychological Well-being, the Satisfaction with Life Scale, Depression Anxiety and Stress Scales, and others. Li and colleagues (2020) introduced the concept of word tagging for retrieving emotional states through an Emotional Recall Task, where participants listed 10 emotional states experienced in the last month. The study demonstrated a significant correlation between the Emotional Recall Task and PANAS, shedding light on diverse mental search strategies, especially in individuals with varying positive or negative affect levels (Li et al., 2020). Building on this, Fatima et al. (2021) developed DASentimental, a semi-supervised machine learning model grounded in the Emotional Recall Task. This innovative system learned to map psychometric measurements of anxiety, stress, and depression to user-generated word sequences, employing mental search strategies rooted in cognitive science and associative knowledge modeling.

The literature overview section now ends by saying that: collectively, these studies emphasize the importance of cognitive information, derived from text-based responses to open-ended questions, in significantly improving artificial intelligence's comprehension of emotional states and complementing the traditionally used rating scale measures. This holds particular relevance for clinical researchers exploring the measurement of psychological states within, for example, unstructured clinical notes that are otherwise undocumented and carry no additional insights in a clinical setting.

In the Participants section - did the authors perform any sort of recognition of human participants against bots? Automatic bots can be relatively frequent on platforms like Prolific.

Our response: reCAPTCHA, a security technology was used to distinguish between human users and automated bots on the Qualtrics survey. This point has been added to the manuscript.

How were the target categories depression, serenity, etc. selected? An explanation about this should enrich the Emotional Autobiographical Memory section in the Methods.

Our response: We added the explanation at the end of the introduction section of the manuscript: Two psychological factors are associated with well-being: life harmony and life satisfaction. Additionally, there are two psychological factors linked to mental health issues: depression and worry. While these factors are theoretically distinct, they often exhibit high correlations when

assessed through rating scales. These four psychological constructs represent a blend of positive psychology and clinical psychology elements, each characterized by a specific emotional valence that can be quantified through the use of standardized rating scales. The selection of these four psychological constructs aligns with previous works from our research group, ensuring consistency and facilitating the comparability of findings across studies. In light of the conceptual and criteria-based distinctions between these two constructs, semantic measures in our study are suggested as a means to differentiate between them more distinctly than traditional rating scales.

How many N/As were featured in the final data?

Our response: Having reviewed the dataset we can confirm that there are no N/As in the data. The narratives underwent a manual review to guarantee the exclusion of any nonsensical data. No whole narratives were removed as a result.

We made minimal alterations, primarily addressing typos and repetitive words (e.g., the - hte; the the ...). Due to the minor nature of these changes, we deemed it not worthwhile to conduct a separate analysis on them.

I understand Figure 2 provides not much more information compared to the in-text descriptions of accuracy. It would be better to replace Figure 2 with a confusion matrix, i.e. a 4x4 heatmap showing correct classifications but also mistakes across the 4 emotional dimensions. This would complement Table 4, which rather contains numbers but results difficult to read, as that Table mixes counts and continuous values without a separation.

Our response: Although we are aware that Figure 2 is a simple representation of the data, we prefer to have one figure that communicates the take-home message of the paper.

We have duly noted your input regarding Table 4 and have subsequently divided it into two separate tables: the heat map, as per your recommendation (now identified as Table 4), and the correlation matrix (now identified as Table 5). This division facilitates a more straightforward examination of the results by avoiding mixing the count and continuous values within a single table.

The explanation for the semantic t-test in word clouds is not clear. Please extend that part or move it to the Methods with an additional formula. The note is clearer but it should be moved to the Methods. It would be more informative to couple the word cloud with a table ranking words and their t values.

Our response: The note under the figure has been moved to the method section and the writing has been made clearer.

In the Results, the author mention performing a 10-fold cross validation but report only values for accuracy without uncertainty margings, e.g. 64%

rather than 63.8 \pm 0.9 %. Please add the standard errors from the 10 folds to the results and comment on them - was the multinomial regression consistent in producing the same accuracy values?

Our response: Standard error has been added to these numbers. They are quite low (i.e., 1.2% and 0.7%).

In the Discussion it would be interesting to mention what would happen if the same person repeated the task over multiple times. Might it be that mood and context alter the word classifications for the very same piece of narratives? Could the current AI pipeline account for alterations provided by current mood and environment?

Our response: The following reasoning in response to this comment has been added to the limitations section of the manuscript. In the present scenarios, it is reasonable to assume that participants might have had different moods from each other and were situated in diverse contexts, and we, researchers, did not have information on these factors. We did not have control over these variables. This aspect presents an intriguing avenue for prospective research, where the manipulation of mood could be explored, and a specialized model could be trained to identify and categorize specific emotions, among other factors. Also, a within subjects study could be another possibility to account for repeated measures.

Typos:

and where the system learns -> and the system learns

scales or languages measures -> scales or language measures

Our response: Thanks, addressed.

2nd Feb 24

Dear Professor Sikström,

Thank you for your patience during the peer-review process. Your manuscript titled "Question-Based Computational Language Approach Outperforms Rating Scales in Quantifying Emotional States" has now been seen by 3 reviewers, and I include their comments at the end of this message. They find your work of interest but raised some important points. We are interested in the possibility of publishing your study in *Communications Psychology*, but would like to consider your responses to these concerns and assess a revised manuscript before we make a final decision on publication.

We therefore invite you to revise and resubmit your manuscript, along with a point-by-point response to the reviewers. Please highlight all changes in the manuscript text file.

Editorially, we consider it important that the revision addresses the remaining concerns of Reviewer 2, in particular, please provide a clearer presentation of the preprocessing steps and adopt consistent terminology throughout.

Furthermore, we ask that you include all preregistered confirmatory hypotheses in the main text of the manuscript. If a hypothesis cannot feasibly be tested the reason for the deviation from the preregistration should be made clear in the main manuscript. The preregistration contains 5 hypotheses. Hypothesis 4 is stated as exploratory and can be excluded, but this needs to be transparently stated. Hypothesis 5 looks like it may be a mistake. Please clarify if this is something you meant to test, and if so please include the tests in the main manuscript. For Hypothesis 3, do estimation scales refer to rating scales, and a comparison of the correlations presented in Table 5? We ask for clarity and transparency regarding the preregistration.

Additionally, all statements or interpretations of the results should be supported by appropriate, fully reported statistics. In detail, frequentist inferential statistics should be reported as follows wherever they occur (main text, Figure captions, Tables, SI): statistic(degrees of freedom) = value, $p = value$, effect size statistic = value, % Confidence Intervals = values. Comparisons of relationships between variables must also be supported by appropriate statistics, rather than rhetoric comparisons.

It is not permissible to interpret the absence of evidence as evidence of absence (in the case of null results). Support for the null hypothesis over the alternative hypothesis cannot therefore be inferred through a null finding using NHST. Instead, appropriate statistical tests (e.g., Bayes Factors or equivalence tests) must form the basis of any interpretation.

I am attaching an Editorial Requests Table that details critical reporting requirements for the revised manuscript. Please attend to each item and ensure your manuscript is fully compliant. We are requesting that your manuscript aligns with these requirements as this facilitates the evaluation of your manuscript, reducing delays in re-review and potential future acceptance. If your revised manuscript is not aligned with these requests on major issues, such as those concerning statistics, it

may be returned to you for further revisions without re-review. Additional information can be found in our style and formatting guide Communications Psychology formatting guide.

Please use the following link to submit your

- revised manuscript,
- point-by-point response to the referees' comments,
- cover letter (as a separate document),
- the Editorial Policy Checklist (see below),
- the Reporting Summary (see below), and
- the completed Editorial Request Table (attached):

[link redacted]

Best regards,

Jonna K. Vuoskoski

Jonna K. Vuoskoski, PhD
Editorial Board Member
Communications Psychology
orcid.org/0000-0003-0049-4373

REVIEWER EXPERTISE:

Reviewer #1: Natural language processing

Reviewer #2: Mental health, Natural language processing

Reviewer #3: Natural language processing, psychometrics

REVIEWER REPORTS:

Reviewer #1 (Remarks to the Author):

The authors have satisfactorily answered the raised concerns. The revised manuscript is sufficiently detailed and has addressed the reviewers' comments.

Reviewer #2 (Remarks to the Author):

Thank you to the authors for taking the time to address my comments so thoroughly! I enjoyed reading the revised version of the manuscript, which I think provides a clearer and more balanced reporting of the study and its findings.

I only have a few additional comments:

1. In response to the authors' removal of analyses described in lines 369-371 of the original version of the manuscript, I would very much recommend that results from any analyses that were completed be reported adequately, even if they do not reflect good performance in detecting the emotional constructs. This is because they may nevertheless be relevant to other researchers who are working within this area, who may wish to complete similar analyses and potentially prevent the duplication of efforts.
2. Related to the revision on page 10, where the authors write "The selection of these four psychological constructs aligns with previous works from our research group, ensuring consistency and facilitating the comparability of findings across studies", it may be useful to cite the authors' prior work being referenced here.
3. The authors write in the revised manuscript for pre-processing that "Descriptive word answers comprising sentences or strings of words rather than one descriptive word in each response box were removed" but earlier under semantic measures, they write "phrases containing more than one word were not allowed and participants were not permitted to proceed in the survey if more than one word was entered in the text field" – so were participants unable to provide responses with > 1 word or were responses with > 1 word removed? In general, some of the revised content related to pre-processing is still unclear (when the authors say "no whole narratives were removed" does this refer to the autobiographical narratives or the descriptive words, or both?)
4. I appreciate that the authors included the content about bias – as there is a concern that rating scales may not capture mental health symptoms and constructs as well in all genders or racial/ethnic groups, which can lead to misdiagnosis and other inequities (e.g., in the case of depression or schizophrenia for black patients) – it's unclear whether this same issue would extend to open-ended or text based responses, given that these have potential to capture a wider range of experiences. That said, an additional question that I am curious about is whether the authors stratified the ML predictions by gender, nationality or education, to see if ML model errors differed based on these factors (which could suggest disparities)? If not, perhaps this could be a point to add in Limitations and Future Directions (apologies this did not come to mind in my prior recommendations!)
5. I would just recommend ensuring that the various responses collected are described consistently – for example, the word responses are variously described as "word-based responses", "language based responses", "descriptive words", "keywords", "semantic measure(s)"...etc. And there is just a minor typo in one of the revised lines on page 32: "and efforts to mitigate social biases such as race,

ethnicity, and religion to make word embedding more neutral (Ahn & Oh, 2021; Bartl et al., 2020; Mozafari et al., 2020)" (should embeddings be plural?)

Reviewer #3 (Remarks to the Author):

The authors cleverly addressed all my points and the manuscript is now suitable for publication in CommPsychol.

EDITORIAL POLICIES

We ask that you ensure your manuscript complies with our editorial policies and reporting requirements.

To that end, we require revised manuscripts to be accompanied by two completed items: a reporting summary that collects information on study design and procedure, and an editorial policy checklist that verifies compliance with all required editorial policies.

- Nature Research Reporting Summary
- Editorial Policy Checklist

All points on the policy checklist must be addressed. Your revised manuscript can only be sent back to the referees if these checklists are completed and uploaded with the revision.

Notes: If you have submitted a Stage 1 Registered Report, Review, Primer, Comment, or Perspective you do not need to submit these forms. If you have already submitted these forms, you may disregard this request.

* TRANSPARENT PEER REVIEW: Communications Psychology uses a transparent peer review system. This means that we publish the editorial decision letters including Reviewers' comments to the authors and the author rebuttal letters online as a supplementary peer review file. However, on author request, confidential information and data can be removed from the published reviewer reports and rebuttal letters prior to publication. If your manuscript has been previously reviewed at another journal, those Reviewers' comments would not form part of the published peer review file.

If you experience problems in linking your ORCID, please contact the Platform Support Helpdesk.

Reviewer #1 (Remarks to the Author):

The authors have satisfactorily answered the raised concerns. The revised manuscript is sufficiently detailed and has addressed the reviewers' comments.

Reviewer #2 (Remarks to the Author):

Thank you to the authors for taking the time to address my comments so thoroughly! I enjoyed reading the revised version of the manuscript, which I think provides a clearer and more balanced reporting of the study and its findings.

I only have a few additional comments:

1. In response to the authors' removal of analyses described in lines 369-371 of the original version of the manuscript, I would very much recommend that results from any analyses that were completed be reported adequately, even if they do not reflect good performance in detecting the emotional constructs. This is because they may nevertheless be relevant to other researchers who are working within this area, who may wish to complete similar analyses and potentially prevent the duplication of efforts.

Our response: We have added the analysis that we removed from the previous submission, showing that basing the categorization on individual rating scales items (N = 26) does not improve over using the total score (N = 4) of these rating scales: "Basing the categorization on individual items of the four rating scales (26 items in total, i.e. 9 items for PHQ-9, 7 for GAD-7, and 5 for SWLS and 5 HILS) did not improve the accuracy in categorization (30%)."

2. Related to the revision on page 10, where the authors write "The selection of these four psychological constructs aligns with previous works from our research group, ensuring consistency and facilitating the comparability of findings across studies", it may be useful to cite the authors' prior work being referenced here.

Our response: The following references were added: O. Kjell et al., 2019; K. Kjell et al., 2021; Sikström et al., 2023.

3. The authors write in the revised manuscript for pre-processing that "Descriptive word answers comprising sentences or strings of words rather than one descriptive word in each response box were removed" but earlier under semantic measures, they write "phrases containing more than one word were not allowed and participants were not permitted to proceed in the survey if more than one word was entered in the text field" – so were participants unable to provide responses with > 1 word or were responses with > 1 word removed? In general, some of the revised content related to pre-processing is still unclear (when the authors say "no whole narratives were removed" does this refer to the autobiographical narratives or the descriptive words, or both?

Our response: That's a valid observation. The confusion regarding the treatment of descriptive word responses containing multiple words might have stemmed from multiple revisions. We have clarified this by removing the statement: "Descriptive word answers comprising sentences or strings of words rather than one descriptive word in each response box were removed." In fact, multiple words in the text box were not allowed, hence no such responses were recorded.

Regarding the remaining pre-processing of semantic data paragraphs, we have provided clarification on the pre-processing of data specifically related to descriptive words and autobiographical narratives. We anticipate that this clarification resolves any confusion.

4. I appreciate that the authors included the content about bias – as there is a concern that rating scales may not capture mental health symptoms and constructs as well in all genders or racial/ethnic groups, which can lead to misdiagnosis and other inequities (e.g., in the case of depression or schizophrenia for black patients) – it's unclear whether this same issue would extend to open-ended or text based responses, given that these have potential to capture a wider range of experiences. That said, an additional question that I am curious about is whether the authors stratified the ML predictions by gender, nationality or education, to see if ML model errors differed based on these factors (which could suggest disparities)? If not, perhaps this could be a point to add in Limitations and Future Directions (apologies this did not come to mind in my prior recommendations!)

Our response: Indeed, we did not categorize the machine learning predictions based on gender or any other demographic variables. While we believe using autobiographical narratives could mitigate bias, it would be interesting to explore this further down the line. We have acknowledged this aspect in the limitations section as follows:

“The present study did not stratify NLP predictions by gender, nationality, or education to investigate possible variations in the resulting model. Recognizing the need for customization to address the diverse dimensions of demographic factors and mental health states, including cognition, physiology, and behavior, could enhance the relevance and effectiveness of the assessment tool in real-world applications. The perpetuation of misdiagnoses and disparities observed with rating scales, particularly evident in situations like depression or schizophrenia diagnoses among black patients, must not be allowed to transfer to open-ended text-based responses analysed using NLP and therefore requires thorough attention.”

5. I would just recommend ensuring that the various responses collected are described consistently – for example, the word responses are variously described as “word-based responses”, “language based responses”, “descriptive words”, “keywords”, “semantic measure(s)”...etc. And there is just a minor typo in one of the revised lines on page 32: "and efforts to mitigate social biases such as race, ethnicity, and religion to make word embedding more neutral (Ahn & Oh, 2021; Bartl et al., 2020; Mozafari et al., 2020)" (should embeddings be plural?)

Our response: Thank you for your observing the typo. That has now been corrected.

We have also addressed the issue regarding the use of different descriptors for some of the concepts. We now use the term descriptive word responses throughout the manuscript (the term "keyword" is retained solely within direct quotations of what participants observed in the survey and the terms 'word-based responses' and 'language based responses' have been removed). We opted to continue using the term 'semantic measures' in some, more general parts of the manuscript, because they encompass a wider umbrella term, including both autobiographical narrative and descriptive word responses. We have made sure to use the term in the correct context, specifically referring to semantic measures as a comprehensive category, rather than exclusively narratives or descriptive word responses.

Reviewer #3 (Remarks to the Author):

The authors cleverly addressed all my points and the manuscript is now suitable for publication in CommPsychol.